# Communication-Efficient Diffusion Denoising Parallelization via Reuse-then-Predict Mechanism

**Kunyun Wang**[*]
School of Computer Science,
Shanghai Jiao Tong University
wkykaixin@sjtu.edu.cn

**Bohan Li**[*]
School of Computer Science,
Shanghai Jiao Tong University
everlastingnight@sjtu.edu.cn

**Kai Yu**
School of Computer Science,
Shanghai Jiao Tong University
kai.yu@sjtu.edu.cn

**Minyi Guo**
School of Computer Science,
Shanghai Jiao Tong University
guo-my@cs.sjtu.edu.cn

**Jieru Zhao**[†]
School of Computer Science,
Shanghai Jiao Tong University
zhao-jieru@sjtu.edu.cn

## Abstract

Diffusion models have emerged as a powerful class of generative models across various modalities, including image, video, and audio synthesis. However, their deployment is often limited by significant inference latency, primarily due to the inherently sequential nature of the denoising process. While existing parallelization strategies attempt to accelerate inference by distributing computation across multiple devices, they typically incur high communication overhead, hindering deployment on commercial hardware. To address this challenge, we propose **ParaStep**, a novel parallelization method based on a reuse-then-predict mechanism that parallelizes diffusion inference by exploiting similarity between adjacent denoising steps. Unlike prior approaches that rely on layer-wise or stage-wise communication, ParaStep employs lightweight, step-wise communication, substantially reducing overhead. ParaStep achieves end-to-end speedups of up to **3.88**× on SVD, **2.43**× on CogVideoX-2b, and **6.56**× on AudioLDM2-large, while maintaining generation quality. These results highlight ParaStep as a scalable and communication-efficient solution for accelerating diffusion inference, particularly in bandwidth-constrained environments.

## 1 Introduction

Benefiting from advances in deep learning and increasingly powerful hardware, diffusion models have demonstrated remarkable performance in image generation [30, 31, 33, 16, 19], video generation [38, 15, 26, 2, 40], and audio generation [35, 11, 8, 24, 25]. However, their widespread adoption is severely limited by substantial inference latency. This issue becomes particularly critical in long-form video generation, where producing a few minutes of video may require hours of GPU computation.

---

[*]Equal contribution.
[†]Corresponding author.

39th Conference on Neural Information Processing Systems (NeurIPS 2025).

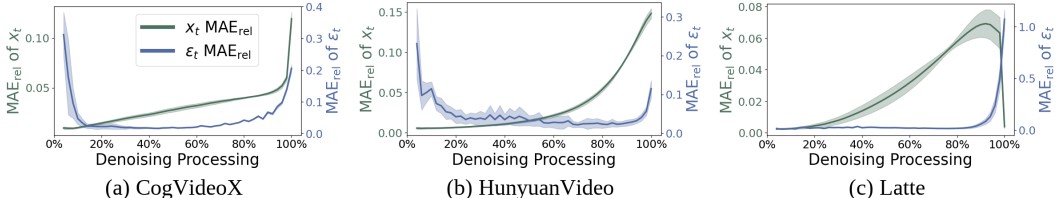

| (a) CogVideoX | (b) HunyuanVideo | (c) Latte |

Figure 1: Relative MAE (as defined in Equation (7)) between adjacent denoising steps $t$ and $t+1$ for the noisy sample $\mathbf{x}_t$ and the predicted noise $\boldsymbol{\epsilon}_t$. Here, 0% on the x-axis indicates the first step of the denoising process, and 100% indicates the last.

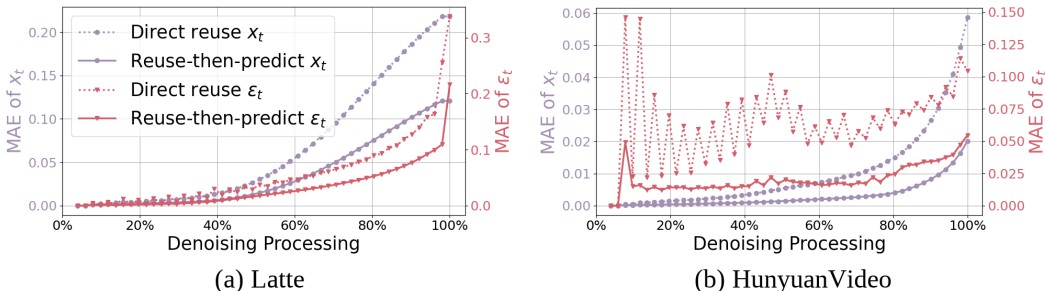

| (a) Latte | (b) HunyuanVideo |

Figure 2: Dotted lines show the difference in noisy sample $\mathbf{x}_t$ between the original model and direct reuse process (stride = 2), while solid lines compare the original model with our reuse-then-predict process (degree = 2). Results for predicted noise $\boldsymbol{\epsilon}_t$ follow the same pattern. Reuse-then-predict process results in smaller deviations from the original model compared to direct reuse process.

The latency primarily stems from two factors: the high computational cost of the noise predictor—typically implemented using architectures such as DiT [29] or U-Net [32]—and the inherently sequential nature of the denoising process, which invokes the noise predictor repeatedly across dozens or even hundreds of timesteps. Notably, in DiT models with 3D attention, this cost scales quadratically with both spatial resolution and temporal length.

To reduce the inference latency of diffusion models, researchers have proposed a variety of solutions. Distillation techniques [27, 34] can reduce the computational and memory overhead of diffusion models. Other works [18, 36] employ post-training quantization (PTQ) to compress full-precision models into 8-bit or 4-bit representations without retraining, thereby reducing computation. Caching-based methods have also been explored to eliminate redundant computation during the denoising process [37, 1, 39, 42]. While effective, these approaches do not leverage the potential of distributed computing, which has become a cornerstone of modern deep learning systems.

To address this, several works have proposed parallelization strategies that leverage distributed infrastructure to accelerate diffusion models [17, 3, 6, 5]. These methods perform operation-wise, layer-wise, or stage-wise communication to exchange essential tensors, which incurs substantial communication cost. Despite their effectiveness, the huge communication overhead makes them impractical outside of high-bandwidth, data center–scale environments. Therefore, there remains a pressing need for parallel approaches with minimal communication requirements to enable practical deployment on commercial hardware.

We analyze the differences in noisy samples and predicted noise across adjacent denoising steps. As illustrated in Figure 1, except for the initial and final few steps, the predicted noises across neighboring steps exhibit high similarity. This observation suggests that it is possible to reuse the noise generated in the previous step to skip the computation of the current step, as shown in Figure 3(b). However, such direct reuse leads to noticeable degradation in generation quality.

To address this issue, we propose a *reuse-then-predict* mechanism. Specifically, the noise from the previous step is reused to generate the noisy sample for the current step, and then a fresh noise prediction is performed based on this sample. Owing to the robustness of noise predictors, the newly predicted noise better approximates the original noise than direct reuse, as evidenced in Figure 2. This procedure can be parallelized through careful scheduling, illustrated in Figure 3(c), enabling acceleration without significant loss in generation quality. Communication is required only once per

denoising step, and only the samples and noises need to be transmitted. This low communication overhead makes ParaStep highly suitable for low-bandwidth environments, such as systems connected via PCIe Gen3. A detailed analysis of the communication pattern is provided in Section 4.2.

Since the difference between adjacent timesteps is large in the early stages, and errors introduced by reuse can accumulate over multiple steps, it is necessary to perform the original denoising process during this phase before switching to the reuse-then-predict mechanism. We refer to these initial steps as *warm-up* steps. The number of warm-up steps is a critical hyperparameter: setting it too low may cause significant performance degradation, while setting it too high can limit the achievable speedup. We will analyze the impact of warm-up step selection in Section 5.5. Although the difference between adjacent timesteps is also large in the final few steps, the errors introduced by reuse only accumulate over a small number of steps and do not influence generation performance. Therefore, we continue to apply the reuse-then-predict mechanism during this phase to maintain acceleration.

For non-compute-intensive models, such as audio diffusion models, distributed computing may be unnecessary. In such cases, enabling ParaStep on a single device can achieve acceleration with lower computational cost. Based on this insight, we develop a single-device variant of ParaStep, called BatchStep, which performs adjacent noise predictions within a single batch rather than distributing them across multiple devices. This design reduces computational overhead while preserving the acceleration benefits of our approach for lightweight models. However, for compute-intensive models such as image and video diffusion, BatchStep is not suitable due to their high resource demands. Our code is available at https://github.com/sjtu-zhao-lab/ParaStep.

We summarize our key contributions as follows:

- We propose a reuse-then-predict mechanism that mitigates the quality degradation caused by direct reuse. Building on this mechanism, we introduce a novel distributed sampling method, termed **ParaStep**, which enables step-wise parallelization across devices for faster inference.
- We analyze the communication characteristics of ParaStep and demonstrate that it achieves significant speedup even under low-bandwidth settings.
- We extend ParaStep to a single-device variant tailored for non-compute-intensive diffusion models, enabling efficient adjacent-step prediction without cross-device communication.
- We perform comprehensive experiments in various diffusion-based models, demonstrating that **ParaStep** is an effective and generalized method applicable to both vision and audio modalities.

## 2   Related works

**Cache**   Several works utilize caching mechanisms to reduce the computational cost of the noise predictor [37, 1]. DeepCache [37] leverages the structural properties of the U-Net architecture, specifically its skip connections between shallow and deep blocks. During less critical timesteps, DeepCache skips the computation of deep blocks by directly forwarding the output of shallow blocks to their corresponding counterparts in deeper layers. In a different direction, [1] explores caching based on prompt similarity. It constructs a cache that stores intermediate features of previously processed prompts, enabling reuse when similar prompts are encountered. While both approaches reduce the computation of the noise predictor, neither leverages the benefits of distributed computation.

**Parallelism**   Conventional parallel strategies such as data parallelism, pipeline parallelism, and tensor parallelism are generally unsuitable for reducing the latency of diffusion models. Data parallelism and pipeline parallelism are primarily designed to improve throughput and provide little benefit for latency reduction. While tensor parallelism is effective for accelerating large language models (LLMs), it is less suitable for diffusion models due to their large activation sizes.

To fully exploit the computational power of distributed GPUs, several methods have been specifically designed to leverage the intrinsic properties of diffusion models for acceleration [17, 3, 6, 5]. Given a parallelism degree of $p$, DistriFusion [17] partitions the input latent of the noise predictor into $p$ patches and employs all-to-all communication to merge them before each attention and convolution operation. PipeFusion [6] reduces communication costs by reusing one-step stale feature maps in patch-level parallelism. AsyncDiff [3] divides the noise predictor into $p$ stages, assigning each stage to a different device and employing pipeline parallelism to achieve speedup. xDiT [5] further combines PipeFusion with Ring Attention [23] to achieve significant acceleration. However, the high

communication overhead associated with these methods restricts their effectiveness to high-bandwidth environments, which are typically found only in expensive data center infrastructures. Moreover, except for Ring Attention [23], all of these approaches rely on approximate parallelism methods, which introduce noticeable deviations in the generated results compared to the original model.

**Reuse**  TeaCache [22] proposes a selective mechanism to reuse the generated noise from the previous timestep, thereby skipping the computation required for the relatively unimportant timesteps. It achieves acceleration with only minor degradation in generation quality. The key difference between TeaCache and our approach lies in their focus: TeaCache addresses the question of *when* to reuse, while our method focuses on *how* to reuse. Specifically, our reuse-then-predict mechanism yields a smaller performance drop by predicting a refined noise estimate rather than directly reusing previous outputs. Moreover, we demonstrate that TeaCache can be seamlessly combined with our method to further improve generation quality while maintaining the same level of acceleration.

## 3 Preliminaries

Diffusion models are a class of generative models that progressively transform data from the original distribution into a Gaussian distribution through a forward diffusion process, and then reconstruct the original data from Gaussian noise via a reverse (backward) denoising process.

**Forward diffusion process**  Given a data sample $\mathbf{x}_0 \sim q(x)$ drawn from the original distribution, Gaussian noise is gradually added over $T$ steps to produce the final sample $\mathbf{x}_T$, which is a pure Gaussian noise. To avoid simulating the full forward trajectory step by step, $\mathbf{x}_t$ can be sampled via:

$$q(\mathbf{x}_t \mid \mathbf{x}_0) = \mathcal{N}\left(\mathbf{x}_t; \sqrt{\bar{\alpha}_t}\mathbf{x}_0, (1 - \bar{\alpha}_t)\mathbf{I}\right) \tag{1}$$

**Backward diffusion process**  Given $\mathbf{x}_t$, the sample $\mathbf{x}_{t-1}$ is drawn from the conditional distribution:

$$p_\theta(\mathbf{x}_{t-1} \mid \mathbf{x}_t) = \mathcal{N}\left(\mathbf{x}_{t-1}; \mu_\theta(\mathbf{x}_t, t), \sigma_t^2\mathbf{I}\right) \tag{2}$$

where $\sigma_t$ is either predefined or learned and the mean $\mu_\theta(\mathbf{x}_t, t)$ is computed as:

$$\mu_\theta(\mathbf{x}_t, t) = \frac{1}{\sqrt{\alpha_t}}\left(\mathbf{x}_t - \frac{1 - \alpha_t}{\sqrt{1 - \bar{\alpha}_t}}\epsilon_t\right) \tag{3}$$

**Denoising computation**  Given a condition $c$, a timestep $t$, and a noisy sample $\mathbf{x}_t$, the noise predictor $\epsilon_\theta$—typically implemented as a U-Net [32] or DiT [29]—estimates the noise $\epsilon_t$. A scheduler then uses this noise to sample $\mathbf{x}_{t-1}$:

$$\epsilon_t = \epsilon_\theta(\mathbf{x}_t, t, c), \quad \mathbf{x}_{t-1} = \textbf{Scheduler}(\mathbf{x}_t, t, \epsilon_t) \tag{4}$$

**Training**  In DDPM [10], model parameters are optimized to minimize the squared error between the true noise and the predicted noise:

$$\nabla_\theta \left\| \epsilon - \epsilon_\theta\left(\sqrt{\bar{\alpha}_t}\mathbf{x}_0 + \sqrt{1 - \bar{\alpha}_t}\epsilon, t\right) \right\|^2 \tag{5}$$

where $\bar{\alpha}_t$ is derived from a predefined noise schedule $\{\beta_t\}$. At each denoising step $t$, the model aims to recover the noise originally added to $\mathbf{x}_0$ to obtain $\mathbf{x}_t$. Since the differences between adjacent noisy samples are often small, the predicted noise $\epsilon_t$ tends to vary only slightly across timesteps.

**Flow Matching**  Flow Matching[21] is a generalization of diffusion[7], sampling using the ordinary differential equation. In Flow Matching, the generative process transforms sample $\mathbf{z}_0$ from a simple reference distribution into sample $\mathbf{z}_1$ from the target distribution by integrating a learned velocity field $\mathbf{v}_\theta(\mathbf{z}, t)$ over time. This process is discretized as:

$$\mathbf{z}_{t+\Delta t} = \mathbf{z}_t + \Delta t \cdot \mathbf{v}_\theta(\mathbf{z}_t, t) \tag{6}$$

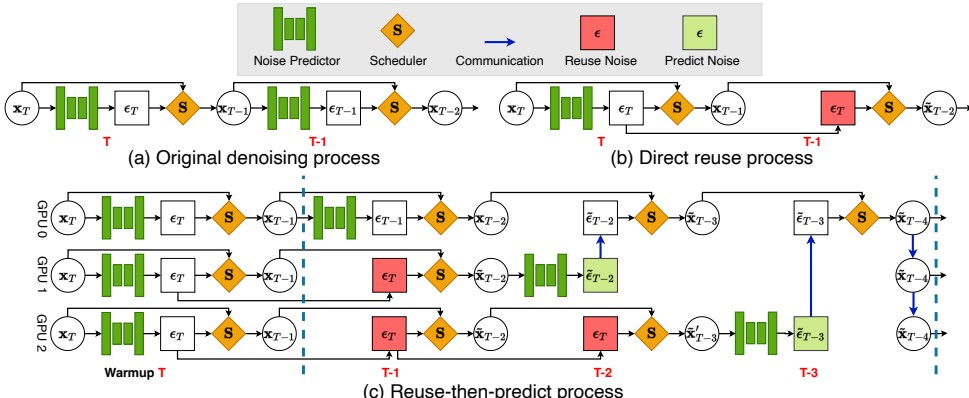

Figure 3: Illustration of the computation process of a diffusion model. (a) The original computation process. (b) Reusing noise $\epsilon_T$ from the previous timestep to skip the computation of noise prediction in the current timestep $T - 1$. (c) ParaStep: adjacent-step noise prediction is distributed across GPUs using reuse-then-predict, enabling parallel denoising with minimal communication. Since the computational cost of scheduler operations is negligible, the noise predictor computations on GPU0, GPU1, and GPU2 are fully parallelized.

## 4   Methods

In conventional diffusion models, the reverse process is implemented as a sequential denoising procedure. The noise predictor $\epsilon_\theta$ is invoked repeatedly to predict the noise $\epsilon_t$, given inputs $\mathbf{x}_t$, $t$, and a conditioning signal $c$. For simplicity, we omit $c$ in the remainder of this paper. The scheduler then uses the predicted noise $\epsilon_t$ to compute the denoised sample $\mathbf{x}_{t-1}$. Since $\epsilon_\theta$ is usually a deep neural network with high computational complexity, whereas the scheduler only performs element-wise additions and multiplications, *the dominant source of computational cost and latency in diffusion models lies in the repeated invocation of the noise predictor*.

To transform the Gaussian noise $\mathbf{x}_T$ into a clean sample $\mathbf{x}_0$, the noise predictor must be called sequentially for tens or even hundreds of timesteps, resulting in significant latency during inference. A naive approach to reduce this cost is to reuse the noise predicted in the previous timestep. However, such direct reuse typically leads to a noticeable degradation in generation quality. To mitigate this degradation while preserving efficiency, we propose a *reuse-then-predict* mechanism, termed **ParaStep**. Our method first reuses the noise from the previous timestep to compute the current noisy sample $\mathbf{x}_t$, and then predicts a refined noise estimate $\epsilon_t$ based on that sample. Through careful design, this procedure allows the prediction of adjacent-step noises to be distributed across multiple devices, effectively parallelizing the most expensive part of the computation. Given a parallelism degree of $p$, the overall workload is divided such that each device only performs $\frac{1}{p}$ of the noise prediction steps. Before introducing the detailed implementation of ParaStep, we first analyze the similarity of predicted noise across adjacent timesteps, and then describe how previously generated noise can be efficiently reused in the denoising process.

### 4.1   Reuse-then-predict mechanism

**Reusing noise**   In each denoising step, the noise predictor aims to estimate the total noise that was added to $\mathbf{x}_0$ to obtain $\mathbf{x}_t$. The predicted noise $\epsilon_t$ remains highly similar across adjacent timesteps, as shown in Figure 1. We use the *Relative Mean Absolute Error (Relative MAE)* to quantify the similarity between noisy samples and predicted noise across steps. The metric is defined as:

$$\mathrm{MAE}_{\mathrm{rel}}(\mathbf{x}_t, \mathbf{x}_{t+1}) = \frac{\frac{1}{n} \sum_{i=1}^{n} |x_{t,i} - x_{t+1,i}|}{\frac{1}{n} \sum_{i=1}^{n} |x_{t,i}|} \tag{7}$$

As illustrated in Figure 1, for models such as CogVideoX [38] and HunyuanVideo [15], the relative MAE between adjacent noise predictions drops below 0.1 after the first 20% of the total steps. This observation suggests that previously predicted noise can be reused to bypass less critical computations.

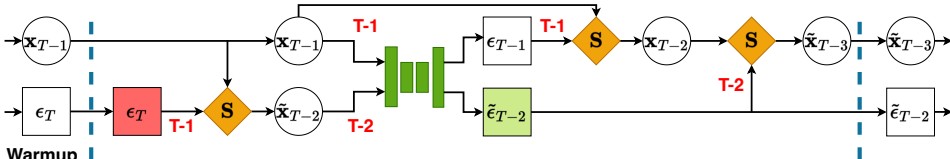

Figure 4: Utilizing batching effect of audio diffusion models, parallel computation of noise predictor can be transformed batching execution on single device.

In Figure 3(a), at timestep $T$, the noise predictor generates $\epsilon_T$ using $\mathbf{x}_T$, after which the scheduler computes $\mathbf{x}_{T-1}$ based on $\epsilon_T$, $\mathbf{x}_T$, and $T$. At timestep $T-1$, the noise predictor estimates $\epsilon_{T-1}$, which is then used to obtain $\mathbf{x}_{T-2}$. In contrast, Figure 3(b) illustrates a reuse strategy: the predictor first estimates $\epsilon_T$ from $\mathbf{x}_T$, which is used to obtain $\mathbf{x}_{T-1}$; then, instead of invoking the predictor again, the model reuses $\epsilon_T$ at timestep $T-1$ to estimate the next noisy sample $\tilde{\mathbf{x}}_{T-2}$. We refer to this strategy as the *direct reuse process*.

Suppose the predicted noise is reused for $s-1$ subsequent timesteps. In that case, the number of predictor invocations is reduced by a factor of $s$, resulting in a corresponding $\frac{1}{s}$ reduction in latency. However, this aggressive reuse introduces a distributional mismatch from the original denoising trajectory, which can lead to significant performance degradation. To mitigate this, we propose a *reuse-then-predict* mechanism that combines reuse with predictive refinement, effectively reducing the quality drop while retaining computational efficiency.

**Parallel denoising via reuse-then-predict mechanism**   As shown in Figure 2, instead of directly reusing the noise predicted in the previous step, a more accurate result can be achieved by first using the reused noise to generate the noisy sample for the current step, and then feeding this sample into the noise predictor to predict the refined noise for the next timestep, we call this mechanism *reuse-then-predict*. Based on this mechanism, we propose a novel distributed sampling method, **ParaStep**, which parallelizes inference by distributing adjacent-step noise prediction across devices.

As illustrated in Figure 3(c), both the noise predictor and the scheduler are replicated on GPU0, GPU1, and GPU2. During the initial warm-up steps, all GPUs follow the standard sequential denoising process. After warm-up, GPU0 predicts $\epsilon_{T-1}$ from $\mathbf{x}_{T-1}$ and timestep $T-1$, and the scheduler then computes $\mathbf{x}_{T-2}$ accordingly. In parallel, GPU1 skips the computation of $\epsilon_{T-1}$ and instead reuses $\epsilon_T$ from the previous step to compute an approximate noisy sample $\tilde{\mathbf{x}}_{T-2}$. It then uses this sample and timestep $T-2$ to predict a refined noise estimate $\tilde{\epsilon}_{T-2}$. Simultaneously, GPU2 performs the same reuse-then-predict process, reusing $\epsilon_T$ twice to compute $\tilde{\mathbf{x}}'_{T-3}$ and then predicting $\tilde{\epsilon}_{T-3}$ from $\tilde{\mathbf{x}}'_{T-3}$ and $T-3$. Since the scheduler introduces negligible computational overhead, the noise prediction steps on GPU1 and GPU2 are effectively parallelized with those on GPU0. Once prediction is complete, GPU1 and GPU2 transmit their predicted noise values, $\tilde{\epsilon}_{T-2}$ and $\tilde{\epsilon}_{T-3}$ respectively, to GPU0. GPU0 then uses these values to compute the next set of noisy samples, $\tilde{\mathbf{x}}_{T-3}$ and $\tilde{\mathbf{x}}_{T-4}$. Finally, the newly generated sample $\tilde{\mathbf{x}}_{T-4}$ is broadcast back to GPU1 and GPU2, and all GPUs advance to timestep $T-4$. This cycle is repeated for subsequent steps, enabling efficient parallelization of adjacent-step noise prediction across multiple devices. We formalize ParaStep with a round-based algorithm, presented in Appendix A.

Assuming a parallelism degree of $p$, ParaStep operates in cycles of length $p$. Each GPU performs one forward pass of the noise predictor per cycle. Since scheduler operations are negligible in cost, noise prediction across all devices is effectively parallelized. Consequently, the total latency of the denoising process is reduced by a factor of $\frac{1}{p}$, achieving a theoretical speedup of $p$.

## 4.2   Communication analysis

To evaluate the communication overhead in multi-GPU environments, we compare the **total** communication volume of three representative methods: AsyncDiff [3] with stage-wise communication, Ring Attention [23] used in xDiT [5] which performs layer-wise communication, and our proposed ParaStep which operates at the step level. For clarity, we assume that the intermediate feature tensors remain of constant size $M$ across all layers of the noise predictor, which is typically implemented using DiT or U-Net. We also assume the model has $L$ attention layers and the parallelism degree is $p$.

**Ring Attention** In each layer, every GPU must gather key and value tensors from all other GPUs. The size of the keys/values from other devices is $\frac{p-1}{p}M$, and the total communication per GPU per layer is therefore $2\frac{p-1}{p}M$. Across all layers, the total communication $\mathcal{C}_{\text{Ring}}$ per step is:

$$\mathcal{C}_{\text{Ring}} = 2L(p-1)M.$$

**AsyncDiff** Each GPU handles a pipeline stage, performing a broadcast of $(p-1)M$ data per step. Since this occurs for all $p$ stages, the total communication $\mathcal{C}_{\text{AsyncDiff}}$ per step is:

$$\mathcal{C}_{\text{AsyncDiff}} = p(p-1)M.$$

**ParaStep** Each cycle (of length $p$) involves $p-1$ send-receive operations and one broadcast. The total communication volume per cycle is $(p-1)M + (p-1)M = 2(p-1)M$. Since one cycle covers $p$ denoising steps, the average communication $\mathcal{C}_{\text{ParaStep}}$ per step is:

$$\mathcal{C}_{\text{ParaStep}} = \frac{2(p-1)M}{p}.$$

These results suggest that *ParaStep significantly reduces communication overhead compared to Ring Attention and AsyncDiff, particularly under limited-bandwidth constraints*. A detailed derivation of the communication analysis is provided in Appendix B.

### 4.3 Leveraging the batching effect for non-compute-intensive models

Image and video diffusion models are typically computationally intensive, where increasing the batch size leads to a linear increase in latency. In contrast, audio diffusion models exhibit a more favorable batching effect. As shown in Figure 5, increasing the batch size results in only a marginal increase in latency. By leveraging this property, we can transform the parallel execution of the noise predictor into a batched inference process on a single device, as illustrated in Figure 4. We refer to this single-device variant as **BatchStep**. The degree of parallelism $p$ in ParaStep corresponds to the cycle length $s$ in BatchStep. This approach achieves a theoretical reduction in per-step computation of approximately $\frac{s-1}{s}$, while completely eliminating the need for multi-device parallelism.

### 4.4 Dynamic degree of parallelism via TeaCache

TeaCache [22] introduces a selective mechanism for reusing previously generated noise, using a larger stride of reuse in less critical timesteps to improve efficiency. We extend ParaStep by integrating the computation schedule of TeaCache. Specifically, the degree of parallelism at each timestep is determined based on the TeaCache schedule. Since ParaStep has low communication overhead, it can achieve similar speedups to Tea-Cache. Moreover, due to the reuse-then-predict mechanism, our method can offer superior generation quality.

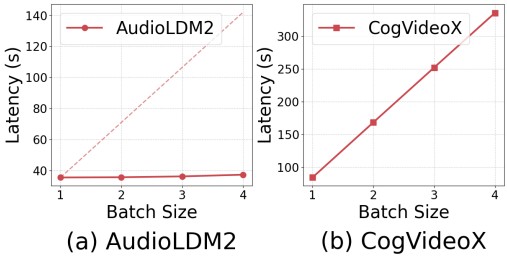

(a) AudioLDM2  (b) CogVideoX

Figure 5: Batching effect of AudioLDM2-large and CogVideoX-2b.

## 5 Experiments

### 5.1 Settings

**Base models and compared methods** Diffusion model is a special case of Flow Matching [21], where both perform a step-by-step computation process to reconstruct the target data distribution. To demonstrate the versatility of our method, we evaluate it on both diffusion models and Flow Matching model. For the Flow Matching model, we adopt Stable Diffusion 3 (SD3) [4] as a representative text-to-image model. For diffusion models, we evaluate on CogVideoX-2b [38] and Latte [26], which are text-to-video models, as well as SVD [2], an image-to-video model. We also include AudioLDM2-large [25] for audio generation. For comparison, we include three state-of-the-art

| Method | Efficiency | | Visual Quality | | | |
|---|---|---|---|---|---|---|
| | Speedup ↑ | Latency (s) ↓ | FID ↓ | LPIPS ↓ | PSNR ↑ | SSIM ↑ |
| SD3 ($T = 50$) | 1 | 18.75 | - | - | - | - |
| AsyncDiff ($p = 2$) | 1.61 | 11.62 | 7.11 | 0.2141 | 16.47 | 0.7290 |
| xDiT-Pipe ($p = 2$) | 1.50 | 12.50 | 6.20 | 0.1943 | 16.68 | 0.7498 |
| **ParaStep** ($p = 2$) | **1.68** | **11.16** | **5.01** | **0.1362** | **18.61** | **0.8157** |

Table 1: Speedup and generation quality on image model SD3, with a resolution of 1440×1440. T is the number of inference steps, and p is the degree of parallelism.

| Method | Latency (s) ↓ | FAD ↓ |
|---|---|---|
| AudioLDM2 ($T = 200$) | 34.80 | 1.6653 |
| ParaStep ($p = 2$) | 18.86 | **1.6651** |
| ParaStep ($p = 4$) | 9.86 | 1.6716 |
| BatchStep ($s = 2$) | 17.74 | 1.6671 |
| BatchStep ($s = 4$) | **9.45** | 1.6699 |

Table 3: Generation latency and FAD on AudioLDM2-large. $p$ is the degree of parallelism, $s$ is the cycle length in BatchStep.

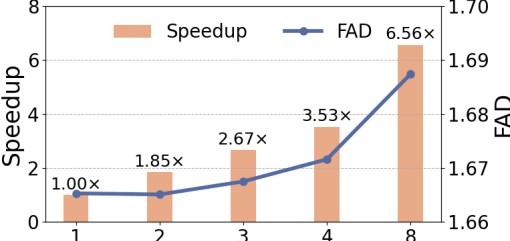

Figure 6: FAD under varying degrees of parallelism using ParaStep. Model: AudioLDM2-large.

parallel methods: AsyncDiff [3], PipeFusion [6], and Ring Attention [23]. The number of warm-up steps for ParaStep is set to 1 for AudioLDM2-large, 5 for SD3, 5 for SVD, 13 for CogVideoX-2b, and 18 for Latte. The number of inference steps is set to 200 for AudioLDM2-large and 50 for all other models.

**Dataset and evaluation metrics**   We use the MS-COCO 2017 [20] validation set for text-to-image model, VBench [12] for text-to-video and image-to-video models, and AudioCaps [14] for audio model, AudioLDM2-large. For text-to-image model, we evaluate performance using Peak Signal-to-Noise Ratio (PSNR), Learned Perceptual Image Patch Similarity (LPIPS) [41], Structural Similarity Index Measure (SSIM), and Fréchet Inception Distance (FID) [9]. For text-to-video and image-to-video models, we use PSNR, LPIPS, SSIM, and VBench (t2v, i2v) score. For audio model, we utilize Fréchet Audio Distance (FAD) [13], which measures the quality of generated audio.

**Hardware**   All experiments are conducted on a machine equipped with 8 NVIDIA 4090 GPUs (24GB each) [28], connected via PCIe Gen3.

## 5.2   Quantitative comparison with baselines

xDiT implements PipeFusion [6] (xDiT-Pipe) and Ring Attention [23] (xDiT-Ring) to accelerate the denoising process. We compare ParaStep with xDiT-Pipe on the image model SD3. Since xDiT does not currently support PipeFusion for video generation models, we instead compare ParaStep with xDiT-Ring for video generation. AsyncDiff is another recent parallel method, and we evaluate ParaStep against AsyncDiff on SVD and SD3. Both xDiT-Pipe, AsyncDiff, and ParaStep rely on approximate methods for parallelization, which introduce varying degrees of deviation from the original model. Although xDiT-Ring is theoretically lossless, its computational logic differs from that of the original model, potentially leading to minor discrepancies in generated outputs. As shown in Table 1 and Table 2, our method achieves state-of-the-art performance compared to baseline methods on both image and video generation models. For SD3, ParaStep demonstrates the highest speedup while maintaining superior generation quality compared to AsyncDiff and xDiT-Pipe. With a parallelism degree of 2, ParaStep achieves a 1.68× speedup over the original denoising process, with a FID of only 5.01. For video generation models, ParaStep also achieves the highest speedup among all baseline methods while maintaining the best generation quality across different degrees of parallelism on SVD and CogVideoX-2b. These results confirm that our method is both highly efficient and introduces minimal impact on generation quality. It is worth noting that, due to the high communication overhead of xDiT-Ring, it provides little to no speedup in our experimental setup.

| Method | Efficiency | | Visual Quality | | | |
|---|---|---|---|---|---|---|
| | Speedup ↑ | Latency (s) ↓ | VBench ↑ | LPIPS ↓ | PSNR ↑ | SSIM ↑ |
| **SVD (14 frames, 1024×576)** | | | | | | |
| SVD ($T = 50$) | 1 | 51.04 | **42.00** | - | - | - |
| AsyncDiff ($p = 2$) | 1.37 | 37.36 | 41.75 | 0.0790 | 25.99 | 0.8366 |
| AsyncDiff ($p = 4$) | 1.80 | 28.43 | 41.56 | 0.1285 | 23.13 | 0.7599 |
| **ParaStep** ($p = 2$) | 1.69 | 30.27 | 41.90 | **0.0278** | **33.35** | **0.9347** |
| **ParaStep** ($p = 4$) | **2.49** | **20.49** | 41.74 | 0.0732 | 26.99 | 0.8433 |
| **Latte (16 frames, 512×512)** | | | | | | |
| Latte ($T = 50$) | 1 | 32.56 | 73.68 | - | - | - |
| xDiT-Ring ($p = 2$) | 1.01 | 32.18 | **73.87** | 0.0424 | 32.79 | **0.9273** |
| xDiT-Ring ($p = 4$) | 1.07 | 30.50 | 73.81 | 0.0431 | **32.80** | 0.9266 |
| **ParaStep** ($p = 2$) | 1.43 | 22.80 | 73.76 | 0.0432 | 32.51 | 0.9258 |
| **ParaStep** ($p = 4$) | **1.82** | **17.91** | 73.76 | 0.0533 | 31.10 | 0.9115 |
| **CogVideoX-2b (45 frames, 512×720)** | | | | | | |
| CogVideoX ($T = 50$) | 1 | 91.89 | 76.95 | - | - | - |
| xDiT-Ring ($p = 2$) | 0.86 | 106.68 | 76.79 | 0.0570 | 32.24 | 0.9284 |
| xDiT-Ring ($p = 4$) | 0.98 | 93.44 | 76.66 | 0.0817 | 29.53 | 0.9028 |
| **ParaStep** ($p = 2$) | 1.47 | 62.50 | **76.97** | **0.0213** | **37.57** | **0.9642** |
| **ParaStep** ($p = 4$) | **1.93** | **47.66** | 76.74 | 0.0359 | 34.34 | 0.9505 |

Table 2: Comparison of speedup and generation quality across four video generation models using different parallel methods.

| Method | Latency (s) ↓ | PSNR ↑ |
|---|---|---|
| TeaCache-slow | 52.25 | 32.25 |
| TeaCache-fast | **36.19** | 24.42 |
| **ParaStep-slow** | 52.61 | **35.76** |
| **ParaStep-fast** | 37.27 | 25.59 |

Table 4: Extend ParaStep by integrating the computation schedule of TeaCache to achieve superior generation quality.

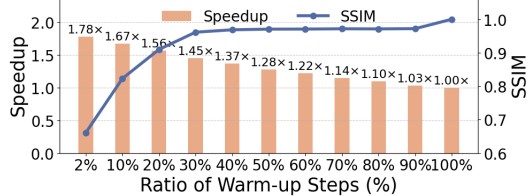

Figure 7: Impact of the number of warm-up steps on the speedup and generation quality of ParaStep. Degree of parallelism: 2. Model: CogVideoX-2b.

## 5.3 Effect of BatchStep

We compare the generation latency and FAD of ParaStep and BatchStep on AudioLDM2-large, with the number of warm-up steps set to 1. As shown in Table 3, both ParaStep and BatchStep achieve acceleration with only a minor degradation in generation quality. Notably, since BatchStep transforms the parallel execution of the noise predictor into a batched inference process on a single device, it eliminates the need for inter-device communication. As a result, BatchStep achieves greater acceleration than ParaStep.

## 5.4 Performance breakdown

To demonstrate that the observed speedup achieved by ParaStep primarily stems from improved communication efficiency, we conducted performance breakdown on the SVD model. Specifically, we measured both the total latency and the communication-related latency for AsyncDiff and our proposed ParaStep. As shown in the Table 5, ParaStep exhibits significantly lower communication latency compared to AsyncDiff, which directly contributes to its higher overall speedup.

| Method | Total (s) | Comp. (s) | Comm. (s) |
|---|---|---|---|
| SVD | 51.04 | 51.04 | - |
| AsyncDiff ($p = 2$) | 39.30 | 30.71 | 8.59 |
| ParaStep ($p = 2$) | 30.52 | 30.42 | 0.10 |
| AsyncDiff ($p = 4$) | 31.30 | 21.01 | 10.29 |
| **ParaStep** ($p = 4$) | **20.08** | 19.94 | 0.14 |

Table 5: Comparison of total latency and breakdown into computation and communication times across different parallel settings.

### 5.5 Ablation studies

**Scalability of our method**    We evaluate the scalability of ParaStep on AudioLDM2-large with a warm-up step count of 1. As shown in Figure 6, ParaStep achieves a 6.56× speedup at a parallelism degree of 8, with only a 0.02 increase in FAD, demonstrating that our method can effectively scale to high degrees of parallelism on AudioLDM2-large. Additional scalability evaluations on other models are provided in Appendix E.2.

**Impact of warm-up steps**    The larger the number of warm-up steps, the fewer the number of parallelizable steps. Due to Amdahl's law, the overall speedup of ParaStep is limited by: $\frac{1}{m+\frac{1-m}{p}}$, where $m$ denotes the ratio of warm-up steps and $p$ represents the degree of parallelism.

However, using too few warm-up steps leads to a degradation in generation quality, as the initial denoising steps are more sensitive to prediction errors. As shown in Figure 7, setting the warm-up ratio to 30% offers a favorable trade-off, achieving strong generation performance with only a moderate reduction in speedup. Increasing the ratio from 30% to 100% yields only marginal improvements in generation quality, but causes a substantial drop in acceleration.

**Extending ParaStep with TeaCache**    By integrating the computation schedule of TeaCache into ParaStep, our method achieves superior generation quality compared to TeaCache, with only a small overhead from communication, as shown in Table 4. Notably, TeaCache-fast adopts a more aggressive reuse schedule than TeaCache-slow. Similarly, ParaStep-slow refers to the integration of the TeaCache-slow schedule into ParaStep, while ParaStep-fast extends ParaStep with the more aggressive TeaCache-fast schedule.

## 6    Conclusion

In this work, we propose ParaStep, a novel parallelization approach for accelerating diffusion model inference. By leveraging a reuse-then-predict mechanism, ParaStep significantly reduces communication overhead while preserving high generation quality. Our approach is grounded in the observation that adjacent denoising steps often exhibit strong similarity, allowing us to bypass costly computations without substantial performance degradation. We further extend this approach with a single-device variant, BatchStep, which transforms parallel execution into efficient batched inference for non-compute-intensive models, such as audio diffusion models. Extensive experimental evaluations confirm that ParaStep delivers superior speedup across diverse modalities, including image, video, and audio generation, while maintaining competitive quality metrics.

## 7    Acknowledgments

This work is sponsored by the National Natural Science Foundation of China (62472273, 62232015).

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

# Appendix

## A    Round-based implementation of ParaStep

---

**Algorithm 1 Round-based implementation of ParaStep**

---

**Require:** Number of inference steps $T$, degree of parallelism $p$, rank of GPU $rank$
**Ensure:** Final output $\mathbf{x}_0$
 1: Initialize noisy sample: $\mathbf{x}_T \sim \mathcal{N}(\mathbf{0}, \mathbf{I})$
 2: Initialize round counter: $round \leftarrow 0$
 3: **for** $t = T, T-1, \ldots, 1$ **do**
 4:     **if** $t \in$ warmup_steps **then** ▷ Perform the original computation process during warm-up steps
 5:         Compute noise: $\epsilon_t = \epsilon_\theta(\mathbf{x}_t, t)$
 6:         Cache noise: $\epsilon_{\text{cache}} \leftarrow \epsilon_t$
 7:         Update sample: $\mathbf{x}_{t-1} = \text{Scheduler}(\mathbf{x}_t, t, \epsilon_t)$
 8:     **else**                           ▷ Perform the ParaStep computation process in non-warm-up steps
 9:         **if** $rank = round$ **then**
10:             Compute noise: $\epsilon_t = \epsilon_\theta(\mathbf{x}_t, t)$
11:             Cache noise: $\epsilon_{\text{cache}} \leftarrow \epsilon_t$
12:             **if** $rank \neq 0$ **then**
13:                 **Send** $\epsilon_t$ to GPU 0
14:             **end if**
15:         **else**
16:             Reuse cached noise: $\epsilon_t \leftarrow \epsilon_{\text{cache}}$
17:             **if** $rank = 0$ **then**
18:                 **Receive** $\epsilon_t$ from GPU $round$
19:             **end if**
20:         **end if**
21:         Update sample: $\mathbf{x}_{t-1} = \text{Scheduler}(\mathbf{x}_t, t, \epsilon_t)$
22:         **if** $round = p - 1$ **then**
23:             **Broadcast** $\mathbf{x}_{t-1}$ from GPU 0 to all GPUs
24:         **end if**
25:         Update round counter: $round \leftarrow (round + 1) \mod p$
26:     **end if**
27: **end for**
28: **return** $\mathbf{x}_0$

---

We formalize ParaStep with a round-based algorithm, presented in Algorithm 1. Each GPU is assigned a unique identifier `rank`, and a variable `round` designates the current master GPU. Assuming a parallelism degree of $p$, ParaStep operates in cycles of length $p$. The master GPU is the one whose `rank` matches the current `round`, and the root GPU is designated as `rank = 0`.

In each round, the master GPU invokes the noise predictor $\epsilon_\theta$ to estimate $\epsilon_t$, while all other GPUs skip this computation. The predicted noise is then sent to the root GPU, which uses it to compute the corresponding noisy sample via the scheduler. In the final round of each cycle (i.e., `round = p − 1`), the root GPU obtains the last noisy sample and broadcasts it to all GPUs.

Each GPU performs one forward pass of the noise predictor per cycle. Since scheduler operations are negligible in cost, noise prediction across all devices is effectively parallelized. Consequently, the total latency of the denoising process is reduced by a factor of $p$, achieving a theoretical speedup of $p$.

## B    Detailed derivation of communication analysis

In this section, we provide a more detailed derivation of the communication volumes for the three parallelism strategies discussed in Section 4.2: **Ring Attention**, **AsyncDiff**, and **ParaStep**. We assume that the intermediate feature tensors remain of constant size $M$ across all layers of the noise predictor, and that the model contains $L$ attention layers. The parallelism degree is denoted as $p$.

### B.1 Ring Attention

In Ring Attention [23], the keys and values on each GPU are split into $p$ partitions, each of size $\frac{1}{p}M$. Before each attention operation, GPUs must gather the keys and values from all other GPUs in a ring-style communication pattern to obtain the complete set of keys and values.

For each GPU, this process involves:

Gathering $p - 1$ partitions, each of size $\frac{1}{p}M$, for both keys and values. - The total communication volume per GPU per layer is therefore:

$$\frac{p-1}{p}M \text{ (keys)} + \frac{p-1}{p}M \text{ (values)} = \frac{2(p-1)}{p}M.$$

Since this communication occurs in every attention layer, the total communication volume per GPU per step is:

$$\mathcal{C}_{\text{Ring, per GPU}} = \frac{2(p-1)}{p}M \cdot L.$$

With $p$ GPUs, the total communication volume across all GPUs per step is:

$$\mathcal{C}_{\text{Ring}} = 2L(p-1)M.$$

### B.2 AsyncDiff

In AsyncDiff [3], the model is divided into $p$ pipeline stages, each assigned to a separate GPU. In each step, every stage (or GPU) must broadcast its outputs to all other GPUs, resulting in a communication volume of:

$$(p-1)M \text{ (per GPU per step)}.$$

Since the model contains $p$ stages, the total communication volume per step across all GPUs is:

$$\mathcal{C}_{\text{AsyncDiff}} = p(p-1)M.$$

### B.3 ParaStep

ParaStep adopts a round-based communication pattern, where each cycle contains $p$ steps. In each cycle:

1. **Noise transfer**: Each GPU (except for GPU 0) sends its predicted noise $\epsilon$ to GPU 0, resulting in:
$$(p-1)M \text{ (noise transfer per cycle)}.$$

2. **Sample broadcast**: In the final step of each cycle, GPU 0 broadcasts the predicted noisy sample $\mathbf{x}$ to all other GPUs, leading to:
$$(p-1)M \text{ (sample broadcast per cycle)}.$$

The total communication volume per cycle is therefore:

$$2(p-1)M.$$

Since one cycle covers $p$ denoising steps, the average communication volume per step is:

$$\mathcal{C}_{\text{ParaStep}} = \frac{2(p-1)M}{p}.$$

## C Limitations

The main limitations of our method are as follows:

Firstly, ParaStep is a parallelization method, which requires additional computational resources to achieve speedup, effectively trading compute for efficiency.

| Method | Speedup ↑ | Latency (s) ↓ | VBench ↑ | LPIPS ↓ | PSNR ↑ | SSIM ↑ |
|---|---|---|---|---|---|---|
| HunyuanVideo (T=50) | 1 | 27.35 | 72.93 | - | - | - |
| xDiT-Ring ($p = 2$) | 1.08 | 25.22 | 72.89 | **0.0071** | **37.04** | **0.9629** |
| xDiT-Ring ($p = 4$) | 1.07 | 25.45 | 72.89 | **0.0071** | **37.04** | **0.9629** |
| **ParaStep** ($p = 2$) | 1.79 | 15.28 | **73.00** | 0.0175 | 34.24 | 0.9482 |
| **ParaStep** ($p = 4$) | **2.67** | **10.25** | 72.77 | 0.0468 | 30.11 | 0.9075 |

Table 6: Speedup and generation quality on HunyuanVideo.

Secondly, ParaStep replicates the entire noise predictor across all GPUs, which means it cannot reduce the per-GPU memory consumption of the noise predictor. A typical diffusion pipeline consists of several components, including a text encoder, a noise predictor (e.g., DiT or U-Net), and a VAE, among others. Among these, the text encoder is often the primary memory bottleneck. Unlike the noise predictor, the text encoder is not replicated; instead, it is partitioned into multiple stages distributed across devices, thereby reducing memory usage on each GPU.

# D    Societal impacts

We propose ParaStep, a novel parallelization approach for accelerating diffusion model inference. ParaStep achieves superior generation quality compared to state-of-the-art parallelization methods, while delivering greater speedup through lightweight, step-wise communication.

ParaStep can be applied in commercial settings to accelerate compute-intensive diffusion models, such as vision-based models, thereby making diffusion technologies more accessible to users and researchers without access to expensive, data center–scale infrastructure. For non-compute-intensive diffusion models, our proposed variant BatchStep enables speedup on a single device, offering a nearly free performance gain with minimal hardware cost.

# E    Supplementary experiments

## E.1    Using ParaStep to accelerate HunyuanVideo

We compare the speedup and generation quality of ParaStep and xDiT-Ring on HunyuanVideo. Since HunyuanVideo requires more GPU memory than the 24GB capacity of an NVIDIA 4090, we apply quantization to reduce memory consumption. In this experiment, each video contains 5 frames with a resolution of 180×180.

As shown in Table 6, ParaStep achieves a 2.67× speedup compared to the original model, with a PSNR of 30.01, indicating that the generated results are highly similar to those of the original model. Notably, when the degree of parallelism is set to 2, the VBench score of ParaStep is even higher than that of the original model.

Although xDiT-Ring is theoretically lossless, its computational logic differs from that of the original model, potentially leading to minor discrepancies in generated outputs. As a result, the generation quality of xDiT-Ring is generally better than that of ParaStep. However, the significant communication overhead of xDiT-Ring severely limits its speedup, achieving only a 1.08× improvement, which is substantially lower than that of ParaStep.

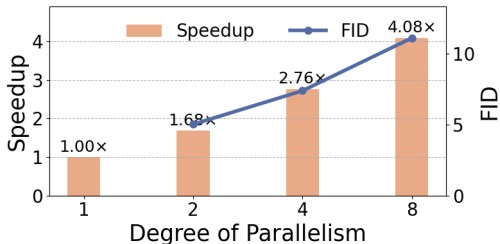
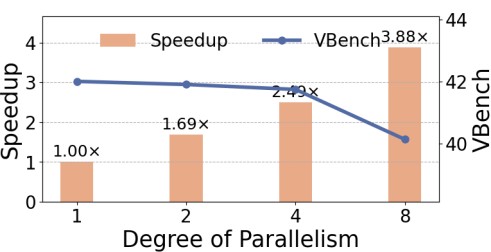

Figure 8: Generation quality under varying degrees of parallelism using ParaStep. Model: SD3.

Figure 9: Generation quality under varying degrees of parallelism using ParaStep. Model: SVD.

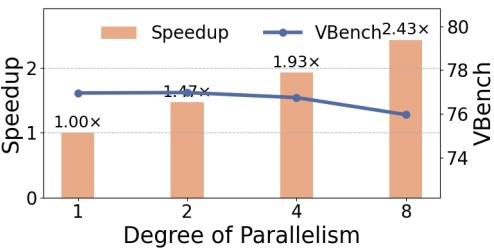
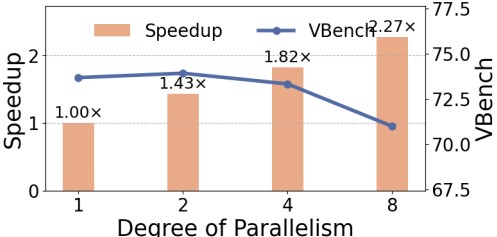

Figure 10: Generation quality under varying degrees of parallelism using ParaStep. Model: CogVideoX-2b.

Figure 11: Generation quality under varying degrees of parallelism using ParaStep. Model: Latte.

### E.2 Scalability of ParaStep on SD3, SVD, CogVideoX-2b, and Latte

We evaluate the speedup and generation quality of ParaStep under varying degrees of parallelism on SD3, SVD, CogVideoX-2b, and Latte. The number of warm-up steps for ParaStep is set to 5 for SD3, 5 for SVD, 13 for CogVideoX-2b, and 18 for Latte, with the number of inference steps fixed at 50. As shown in Figure 8 and Figure 9, with a parallelism degree of 8, ParaStep achieves a 4.08× speedup on SD3 with a FID of only 11.08, and a 3.88× speedup on SVD with a minor VBench score drop of less than 2. However, for CogVideoX-2b and Latte, the speedup is less significant, as shown in Figure 10 and Figure 11, due to the larger number of warm-up steps required for these models.

### E.3 Comparison with single-device efficient inference approach

Single-device efficiency methods such as DeepCache can reduce the latency of diffusion models without requiring additional computational resources. However, directly caching used in DeepCache may cause a significant degradation in generation quality. In contrast, our ParaStep, based on the Reuse-then-Predict mechanism, offers a balanced trade-off by leveraging additional computational resources to achieve higher speedups with minimal quality loss.

To assess the efficiency of ParaStep, we compare it against both DeepCache and AsyncDiff on the SVD model, which adopts the deterministic *EulerDiscreteScheduler*. We denote the cache interval in DeepCache as $s$, and the degree of parallelism in AsyncDiff and ParaStep as $p$. As shown in Table 7, ParaStep consistently outperforms both baselines in terms of generation quality and latency.

### E.4 Visualization

We visualize the outputs of AsyncDiff, xDiT-Pipe, and ParaStep on SD3. The number of inference steps is set to 50, with a resolution of 1440×1440. As shown in Figure 12, ParaStep demonstrates superior generation quality compared to AsyncDiff and xDiT-Pipe.

In the first row of Figure 12, compared to the original diffusion model, AsyncDiff generates an extra hand, while xDiT-Pipe fails to generate a fork and a water cup. In the second row, the logo in the right corner of the bus generated by AsyncDiff differs from that of the original model, and the bag

| Method | Latency (s) ↓ | LPIPS ↓ | PSNR ↑ | SSIM ↑ |
|---|---|---|---|---|
| SVD ($T = 50$) | 52.23 | - | - | - |
| DeepCache ($s = 2$) | 36.42 | 0.0399 | 32.7247 | 0.9155 |
| DeepCache ($s = 4$) | 26.90 | 0.0847 | 26.4912 | 0.8415 |
| DeepCache ($s = 6$) | 23.75 | 0.1397 | 23.1894 | 0.7728 |
| DeepCache ($s = 8$) | 22.24 | 0.1973 | 19.9736 | 0.7160 |
| AsyncDiff ($p = 2$) | 39.28 | 0.0861 | 24.0769 | 0.8328 |
| AsyncDiff ($p = 4$) | 29.65 | 0.1306 | 21.1599 | 0.7505 |
| **ParaStep** ($p = 2$) | 30.61 | **0.0283** | **32.8624** | **0.9358** |
| **ParaStep** ($p = 4$) | **20.36** | 0.0751 | 24.9949 | 0.8448 |

Table 7: Comparison of latency and generation quality across different caching and parallelization strategies.

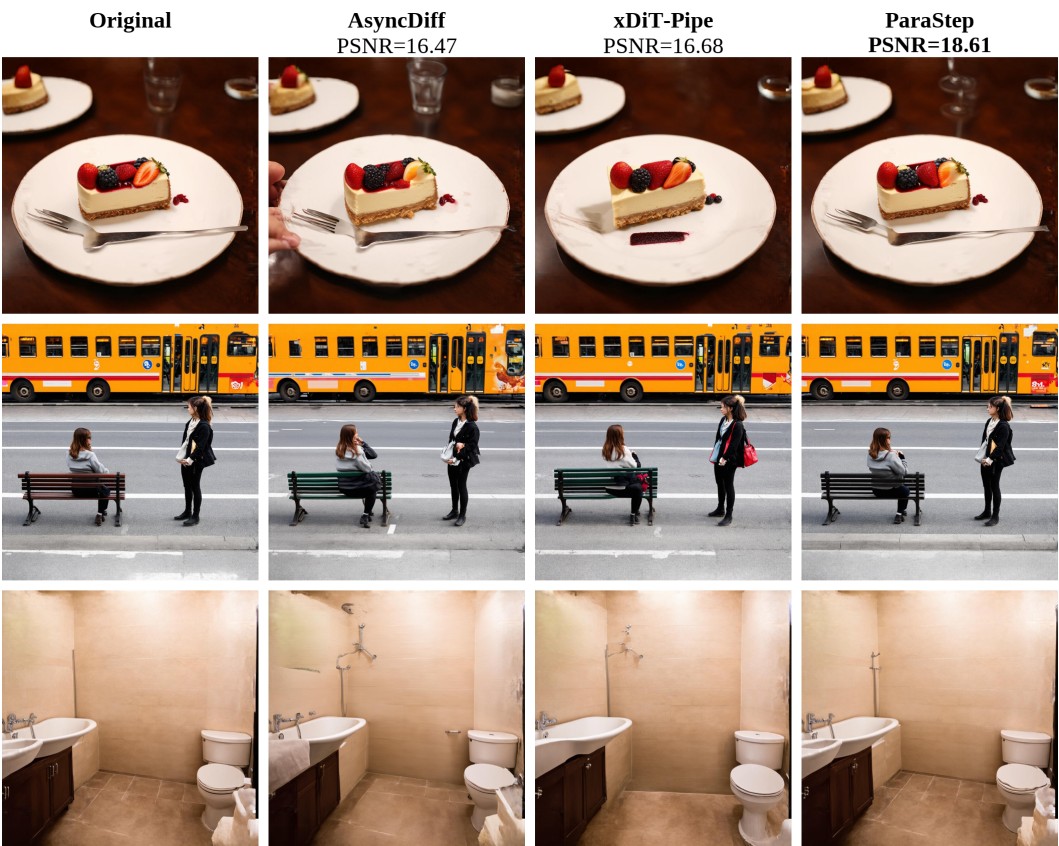

Figure 12: Comparison of generation quality on SD3 using competing methods. PSNR is used to measure the similarity to the original model, with higher values indicating better quality.

generated by xDiT-Pipe is red, which is inconsistent with the original output. In the third row, both AsyncDiff and xDiT-Pipe hallucinate an additional shower that does not appear in the original result.

In contrast, ParaStep achieves higher consistency with the original diffusion model across all examples.

