# OpenReview forum: "Communication-Efficient Diffusion Denoising Parallelization via Reuse-then-Predict Mechanism"
_NeurIPS.cc/2025/Conference — NeurIPS 2025 poster_

### Official Review · Reviewer_eKuX · 2025-07-02

**Clarity:** 3
**Significance:** 3
**Originality:** 2
**Rating:** 4
**Confidence:** 5

**Summary:**

This paper proposes ParaStep, a method to parallelize diffusion model inference by exploiting similarities between adjacent denoising steps. It adopts a reuse-then-predict mechanism, enabling multiple denoising steps to be computed in parallel. For audio diffusion models, where GPU utilization is typically low, the authors instead increase the batch size to process multiple timesteps within a single batch, improving GPU utilization. Experiments on SVD, CogVideoX, and AudioLDM2 demonstrate the effectiveness of the proposed approach.

**Questions:**

- How does ParaStep perform on FLUX.1, the state-of-the-art image generation model?
- What is the performance of ParaStep on diffusion models with fewer steps (e.g., 10 or 20)? Current experiments use 50 steps, which is not a typical setting in practice.

**Ethical Concerns:**

["NO or VERY MINOR ethics concerns only"]

**Final Justification:**

I have read the rebuttal, and most of my concerns were addressed. Therefore, I raised my score to 4.

**Limitations:**

Yes.

**Paper Formatting Concerns:**

No.

**Quality:**

2

**Strengths And Weaknesses:**

## Strengths

* The topic of distributed inference of diffusion models is underexplored but significant.
* The idea is simple, and the paper is easy to follow.
* Using batching to improve GPU utilization for audio diffusion models is interesting.
* The authors provide code to reproduce their results.

## Weaknesses

* The proposed method has limited scalability. As the degree of parallelism increases, staleness is also increased, which degrades generation quality.
* The proposed dynamic degree of parallelism seems not useful. In real-world deployments, the number of devices is typically fixed. If the dynamic degree of parallelism is lower than the number of devices, some compute resources are left idle, which is wasteful.
* Unlike many distributed inference strategies, ParaStep replicates the full model on each device, failing to reduce per-device memory usage.
* While ParaStep has lower communication amounts than methods like Ring Attention and AsyncDiff, those methods hide communication latency via asynchrony. This critical aspect is not mentioned in the paper, which is not fair.
* Leveraging similarities between adjacent denoising steps to accelerate or parallelize diffusion models has been extensively explored in prior work, including DistriFusion, AsyncDiff, DeepCache, and TeaCache. The concept of warmup is also explored in DistriFusion.

---

> ### Author Rebuttal · Authors · 2025-07-27
>
> Dear reviewer eKuX,
>
> Thank you for providing constructive comments. Below, we address each point raised.
>
> **W1: The proposed method has limited scalability. As the degree of parallelism increases, staleness is also increased.**
>
> **Reply:** Thank you for your comment. Similar to AsyncDiff and DeepCache, our ParaStep leverages the similarity across adjacent steps in the diffusion process. As the degree of parallelism increases (in ParaStep or AsyncDiff) or as the cache interval increases (in DeepCache), the generation quality of these methods may degrade. This is a limitation of techniques that rely on inter-step similarity.
>
> However, we propose **reuse-then-predict** mechanism to effectively reduces the staleness caused by Direct Reuse. As a result, ParaStep achieves **higher generation quality** than the sota methods, as shown in Table 2 of our paper. Moreover, since diffusion models typically use no more than 50 inference steps, a parallelism degree of 4 or 8 is generally sufficient to achieve meaningful acceleration. As demonstrated in Figure 6, ParaStep delivers significant speedup with *p* = 8, while maintaining an acceptable quality drop.
>
> **W2: The proposed dynamic degree of parallelism seems not useful.  Some compute resources are left idle, which is wasteful.**
>
> **Reply:** Thank you for your comment. This dynamic parallelism is particularly suitable for **diffusion serving systems**, where the goal is to respond to multiple requests efficiently and with low latency. For example, the system may initially assign 8 devices to a single request, allowing ParaStep to perform the first 16 denoising steps with a parallelism degree of 8. The system can then **reduce the degree** to 4 for subsequent steps, freeing up 4 devices to handle other incoming requests. If the parallelism degree is later increased back to 8 for the original request, only intermediate states—such as the noise ($\epsilon$) and the current sample ($X$)—need to be transferred to the newly assigned devices. The associated communication overhead is minimal.
>
> **W3: Unlike many distributed inference strategies, ParaStep replicates the full model on each device.**
>
> **Reply:** Thank you for your comment. We apologize for this unclear description in Section Limitations. A typical diffusion pipeline consists of several components, including a text encoder, a noise predictor (e.g., DiT or U-Net), and a VAE, among others. The model as mentioned in the paper denotes the noise predictor, and ParaStep only requires replicating the entire noise predictor across all devices. However, the text encoder, typically the primary memory bottleneck, is not replicated. Instead we split the text encoder into multiple stages across devices to reduce per-device memory consumption. We will clarify this point more explicitly in the revised version of the paper. The table below presents the memory breakdown for SD3, CogVideoX-2B, and Latte.
>
> | Pipeline     | Total(MB) | Text encoder(MB) | Noise predictor(MB) |
> | ------------ | --------- | ---------------- | ------------------- |
> | SD3          | 16764     | 11004            | 4024                |
> | CogVdieoX-2b | 14769     | 11004            | 3342                |
> | Latte        | 13188     | 11004            | 2019                |
>
> One may worry that splitting the text encoder increases latency. Since the text encoder is only invoked once to process the input prompt, and splitting it only introduces a single communication step between adjacent stages, the **additional latency incurred is minimal**. To validate this, we conducted experiments on SD3. In the results, **"not split"** refers to using ParaStep without splitting the text encoder, while **"split"** means ParaStep with encoder splitting.
>
> | SD3            | Latency(s) | Memory usage per GPU(MB) |
> | -------------- | ---------- | ------------------------ |
> | p=1            | 18.81      | 16764                    |
> | p=2, not split | 11.16      | 16764                    |
> | p=2, split     | 11.20      | 11386                    |
> | p=4, not split | 6.79       | 16764                    |
> | p=4, split     | 6.84       | 8698                     |
>
> **W4: Methods like Ring Attention and AsyncDiff hide communication latency via asynchrony. This critical aspect is not mentioned in the paper, which is not fair.**
>
> **Reply:** Thank you for your comment. For Ring-Attention, it leverages asynchronous communication to overlap communication with adjacent computation. However, due to the large communication volume and limited memory bandwidth, communication cannot be fully overlapped with computation, which significantly limits the speedup achieved by Ring-Attention. As shown in Table 2 of our paper, Ring-Attention achieves only a 1.07× speedup due to this communication bottleneck, whereas our ParaStep attains a 1.82× speedup under the same configuration.
>
> For AsyncDiff, to the best of our knowledge, AsyncDiff performs synchronous communication. To demonstrate that the speedup achieved by ParaStep primarily stems from improved communication efficiency, we conducted experiments on the SVD model.
>
> |                | Total Latency(s) | Latency of communication(s) |
> | -------------- | ---------------- | --------------------------- |
> | SVD            | 51.04            | -                           |
> | AsyncDiff(p=2) | 39.30            | 8.59                        |
> | ParaStep(p=2)  | 30.52            | 0.10                        |
> | AsyncDiff(p=4) | 31.30            | 10.29                       |
> | ParaStep(p=4)  | **20.08**        | 0.14                    |
>
> **W5: Leveraging similarities between adjacent denoising steps to accelerate or parallelize diffusion models has been extensively explored in prior work, including DistriFusion, AsyncDiff, DeepCache, and TeaCache. The concept of warmup is also explored in DistriFusion.**
>
> **Reply:** Thank you for your comment. The core difference between our ParaStep and the methods you mentioned lies in the **reuse-then-predict** mechanism we propose and adopt. This mechanism is designed to **refine the quality degradation** caused by direct reuse, leading to superior generation quality compared to sota acceleration methods.
>
> To validate this, we conducted experiments comparing ParaStep with both DeepCache and AsyncDiff. As shown, ParaStep outperforms DeepCache and AsyncDiff. Specifically, with a parallelism degree of *p* = 2, ParaStep achieves higher generation quality than both baselines. Furthermore, ParaStep with *p* = 4 not only achieves lower latency than DeepCache with stride *s* = 8, but also delivers higher generation quality, demonstrating its effectiveness in balancing speed and quality.
>
> |                | Latency(s)↓ | lpips↓     | PSNR↑       | SSIM↑      |
> | -------------- | ----------- | ---------- | ----------- | ---------- |
> | SVD(T=50)      | 52.23       | -          | -           | -          |
> | DeepCache(s=2) | 36.42       | 0.0399     | 32.7247     | 0.9155     |
> | DeepCache(s=4) | 26.899      | 0.0847     | 26.4912     | 0.8415     |
> | DeepCache(s=6) | 23.745      | 0.1397     | 23.1894     | 0.7728     |
> | DeepCache(s=8) | 22.237      | 0.1973     | 19.9736     | 0.7160     |
> | AsydnDiff(p=2) | 39.278      | 0.0861     | 24.0769     | 0.8328     |
> | AsydnDiff(p=4) | 29.65       | 0.1306     | 21.1599     | 0.7505     |
> | ParaStep(p=2)  | 30.61       | **0.0283** | **32.8624** | **0.9358** |
> | ParaStep(p=4)  | **20.357**  | 0.0751     | 24.9949     | 0.8448     |
>
> **Q1: How does ParaStep perform on FLUX.1, the state-of-the-art image generation model?** and
>
> **Q2: What is the performance of ParaStep on diffusion models with fewer steps (e.g., 10 or 20)? Current experiments use 50 steps, which is not a typical setting in practice.**
>
> **Reply:** Thank you for your questions. Most of our experiments are conducted under the 50-step setting, as this is the default configuration for models such as CogVideoX-2B and Latte. Setting the number of steps of CogVideoX-2B to 35 results in a significant drop in generation quality.
>
> |                     | Latency(s)↓ | lpips↓     | PSNR↑       | SSIM↑      |
> | ------------------- | ----------- | ---------- | ----------- | ---------- |
> | CogVideoX-2B(T=50)  | 86.92       | -          | -           | -          |
> | CogVideoX-2B(T=40)  | 70.94       | 0.3102     | 18.4771     | 0.6931     |
> | CogVideoX-2B(T=35)  | 64.12       | 0.4033     | 16.2626     | 0.6403     |
> | ParaStep(T=50, p=2) | **58.24**   | **0.0247** | **35.2605** | **0.9588** |
>
> To assess the performance of ParaStep in few-step settings, we conducted experiments on **Flux1.dev**, which supports few-step generation. Here, *p* denotes the degree of parallelism in ParaStep, and *s* represents the stride used in the Direct Reuse method. The results in the table demonstrate that ParaStep achieves speedup under few-step settings, while also outperforming both Direct Reuse and reduced-step generation in terms of output quality.
>
> |                         | Latency(s)↓ | FID↓       | PSNR↑     | SSIM↑      |
> | ----------------------- | ----------- | ---------- | --------- | ---------- |
> | Flux(T=10)              | 5.79        | -          | -         | -          |
> | **Group 1**             |             |            |           |            |
> | Direct Reuse(T=10, s=2) | 3.33        | 157.34     | 17.56     | 0.6427     |
> | Flux(reduced-step, T=5) | 3.30        | 157.44     | 17.56     | 0.6421     |
> | ParaStep(T=10, p=2)     | 3.47        | **141.28** | **18.72** | **0.6751** |
> | **Group 2**             |             |            |           |            |
> | Direct Reuse(T=10, s=4) | 2.50        | 291.83     | 15.69     | 0.5849     |
> | Flux(reduced-step, T=3) | 2.24        | 268.41     | 15.91     | 0.5877     |
> | ParaStep(T=10, p=4)     | 2.60        | 256.70     | 17.36     | 0.6352     |

---

> > ### Comment · Reviewer_eKuX · 2025-08-03
> > **Re: Rebuttal**
> >
> > I have read the rebuttal, and most of my concerns were addressed. However, I disagree with the authors’ response to W3. They only report the memory usage of small diffusion models. To my knowledge, FLUX’s diffusion models have around 12B parameters, SD3.5-Large’s transformer has 8B, and Wan2.1’s transformer has 14B. In these cases, the diffusion model is significantly larger than both the text encoder and the VAE, which is the primary memory bottleneck.

---

> > > ### Author Response · Authors · 2025-08-03
> > > **Response to reviewer eKuX**
> > >
> > > Thank you for your follow-up comment. We agree that in some cases, the diffusion model can indeed be larger than the text encoder. While reducing the memory footprint of diffusion models is important, **the core contribution of our work lies in designing a communication-efficient parallelism strategy to reduce the high latency of diffusion inference**. ParaStep parallelizes computation across diffusion steps to achieve communication-efficient parallelism, which directly contributes to the observed speedup.
> > >
> > > ParaStep does not reduce the memory consumption of the noise predictor (nor does it increase it); similar to data parallelism and sequence parallelism, ParaStep replicates the noise predictor across all devices. However, by splitting the text encoder, our approach reduces its memory usage. In models where the text encoder is the primary memory bottleneck—such as CogVideoX-2B, Latte, SD3, and PixArt-alpha—this design significantly alleviates memory pressure. These models are widely used on commercial hardware, which is the **target deployment setting of our work**. In contrast, models like FLUX, SD3.5-Large, and Wan2.1 are more powerful but are typically deployed in data center–scale environments due to their high memory and computational demands.
> > >
> > > That said, in scenarios where the noise predictor becomes the dominant component, further memory reduction is beneficial. One possible way to reduce memory consumption is through tensor parallelism or pipeline parallelism. However, tensor parallelism incurs substantial communication overhead, making it unsuitable for our bandwidth-limited deployment setting. Pipeline parallelism cannot be directly applied either; AsyncDiff, a variant of pipeline parallelism, is able to split the noise predictor into stages and achieve speedup. However, as shown in Table 2 of our paper, AsyncDiff achieves both lower speedup and lower generation quality compared to our ParaStep.
> > >
> > > Achieving both speedup and memory reduction while maintaining generation quality is particularly challenging under communication-constrained commercial hardware. ParaStep delivers strong speedup and even memory savings in such settings, while preserving high generation quality.
> > >
> > > Quantization offers an effective way to reduce memory consumption by compressing model weights. We believe combining ParaStep with quantization is a promising direction to jointly improve both latency and memory efficiency for large-scale diffusion models. Exploring how to balance memory savings with the potential quality degradation introduced by quantization remains an important and interesting research challenge.

---

> > > > ### Comment · Reviewer_eKuX · 2025-08-04
> > > > **Re: Author Response**
> > > >
> > > > Thanks for the reply. Though the method cannot shard the model, I believe it is still useful in certain cases. I will raise my score to 4.

---

> > > > > ### Author Response · Authors · 2025-08-05
> > > > > **Response to Reviewer eKuX**
> > > > >
> > > > > Thank you very much for rasing the score! We sincerely appreciate your insightful and constructive comments, which have been highly valuable in improving our paper. We will carefully revise the manuscript based on your feedback.

---

### Official Review · Reviewer_gLa1 · 2025-07-02

**Clarity:** 3
**Significance:** 2
**Originality:** 2
**Rating:** 4
**Confidence:** 3

**Summary:**

This paper introduces ParaStep, a communication-efficient parallelization strategy for accelerating inference in diffusion models. ParaStep exploits the high similarity between noises predicted at adjacent denoising steps quantified by relative MAE to enable efficient reuse of previous noise predictions. Instead of sequentially predicting noise at every step, ParaStep first reuses the noise from the prior step to estimate the current noisy sample, then predicts a refined noise for that sample. This “reuse-then-predict” approach is distributed across multiple GPUs, which greatly reduces communication overhead compared to existing layer-wise or stage-wise parallel diffusion methods. The authors demonstrate that ParaStep achieves significant inference speedups while maintaining generation quality. They also introduce BatchStep, a single-device batching variant for lightweight models, and demonstrate the generality of their approach across image, video, and audio diffusion models through comprehensive experiments.

**Questions:**

1. Can the selection of key hyperparameters (e.g., warm-up steps, parallelism degree) be automated or made adaptive across different diffusion models?

2. For single-GPU scenarios, how does ParaStep (or BatchStep) compare to step-reduction methods (e.g., DDIM or consistency models) in terms of speedup and quality, do these approaches outperform ParaStep/BatchStep in practice?

3. Can ParaStep be effectively combined with single-device acceleration methods (such as DeepCache or DDIM) for further speedup, and have the authors benchmarked or analyzed such hybrid approaches?

**Ethical Concerns:**

["NO or VERY MINOR ethics concerns only"]

**Final Justification:**

I have read the rebuttal and I decided to increase my score from 3 to 4.
The paper would benefit from more systematic clarification of the experimental settings and hyper-parameter configurations, but I think the paper propose a pipeline that could be utilized for the deployment of not-that-larger diffusion model series.

**Limitations:**

Yes.

**Paper Formatting Concerns:**

No.

**Quality:**

2

**Strengths And Weaknesses:**

Strength

1. Substantially reduces multi-GPU communication that makes parallel diffusion inference feasible even on bandwidth-limited hardware.

2. Strong compatibility that works across image, video, and audio diffusion models without retraining.

3. Avoids the generation quality degradation typical in naive distributed inference methods benefited from its reuse-then-predict approach.

4. Achieves significant speedup with minimal output quality loss, validated against multiple state-of-the-art parallel baselines.

Weakness

1. Lacks comparison with single-device efficient inference approaches (e.g., DeepCache, DDIM) or their combinations.

2. Requires manual tuning of hyperparameters (warm-up steps, parallelism degree), with limited discussion of robustness or automation.

3. Needs full model replication on each GPU, limiting scalability for very large models.

---

> ### Author Rebuttal · Authors · 2025-07-27
>
> Dear reviewer gLa1,
>
> Thank you for providing constructive comments and valuable suggestions. Below, we address each point raised.
>
> **W1:  Lacks comparison with single-device efficient inference approaches.**
>
> **Reply:** Thank you for your valuable comments. DDIM is a deterministic scheduler that, compared to stochastic schedulers such as DDPM, reduces sampling steps. Since ParaStep is not a scheduler-level work, it is fully compatible with DDIM and other deterministic schedulers. We demonstrate this compatibility in the table below.
>
> To evaluate the efficiency of ParaStep, we compare it against both DeepCache and AsyncDiff. The experiments are conducted on **SVD**, which uses a **deterministic** scheduler EulerDiscreteScheduler. We denote the cache interval in DeepCache as *s* and the degree of parallelism in AsyncDiff and ParaStep as *p*.
>
> |                | Latency(s)↓ | lpips↓     | PSNR↑       | SSIM↑      |
> | -------------- | ----------- | ---------- | ----------- | ---------- |
> | SVD(T=50)      | 52.23       | -          | -           | -          |
> | DeepCache(s=2) | 36.42       | 0.0399     | 32.7247     | 0.9155     |
> | DeepCache(s=4) | 26.899      | 0.0847     | 26.4912     | 0.8415     |
> | DeepCache(s=6) | 23.745      | 0.1397     | 23.1894     | 0.7728     |
> | DeepCache(s=8) | 22.237      | 0.1973     | 19.9736     | 0.7160     |
> | AsydnDiff(p=2) | 39.278      | 0.0861     | 24.0769     | 0.8328     |
> | AsydnDiff(p=4) | 29.65       | 0.1306     | 21.1599     | 0.7505     |
> | ParaStep(p=2)  | 30.61       | **0.0283** | **32.8624** | **0.9358** |
> | ParaStep(p=4)  | **20.357**  | 0.0751     | 24.9949     | 0.8448     |
>
> First, SVD adopts **EulerDiscreteScheduler** as its scheduler. Our method achieves significant acceleration on SVD demonstrates its compatibility with deterministic schedulers. Second, **ParaStep outperforms both DeepCache and AsyncDiff** in terms of generation quality and latency.
>
> **W2: Requires manual tuning of hyperparameters (warm-up steps, parallelism degree), with limited discussion of robustness or automation.**
>
> **Reply:** Thank you for your valuable comment. Regarding the choice of warm-up steps, we find in practice that **10% of the total steps is sufficient for simpler models** such as SVD and SD3, while **30% is a reasonable choice for more complex models** like CogVideoX and Latte. Since generation quality and speedup are both highly sensitive to the number of warm-up steps, but remain stable across different input samples, we can efficiently **profile** a small set of candidates—e.g., [1, 10% × *T*, 20% × *T*, 30% × *T*]—to identify a good trade-off between quality and efficiency.
>
> As for the degree of parallelism, diffusion models typically use no more than 50 inference steps, making **parallelism degrees below 8** practical for achieving speedup. As shown in Figure 6 of our paper, ParaStep achieves significant acceleration with *p* = 8 while maintaining acceptable generation quality. Therefore, in practice, we can simply set the parallelism degree to the number of available devices on the machine (≤ 8).
>
> **W3: Needs full model replication on each GPU, limiting scalability for very large models.**
>
> **Reply:** Thank you for your valuable comment. We apologize for this unclear description in Section Limitations. A typical diffusion pipeline consists of several components, including a text encoder, a noise predictor (e.g., DiT or U-Net), and a VAE, among others. The model as mentioned in the paper denotes the noise predictor, and ParaStep only requires replicating the entire noise predictor across all devices. However, the text encoder, typically the primary memory bottleneck, is not replicated. Instead we split the text encoder into multiple stages across devices to reduce per-device memory consumption. We will clarify this point more explicitly in the revised version of the paper. The table below presents the memory breakdown for SD3, CogVideoX-2B, and Latte.
>
> | Pipeline     | Total(MB) | Text encoder(MB) | Noise predictor(MB) |
> | ------------ | --------- | ---------------- | ------------------- |
> | SD3          | 16764     | 11004            | 4024                |
> | CogVdieoX-2b | 14769     | 11004            | 3342                |
> | Latte        | 13188     | 11004            | 2019                |
>
> One may worry that splitting the text encoder increases latency. Since the text encoder is only invoked once to process the prompt, and splitting it only introduces a single communication step between adjacent stages, the **additional latency incurred is minimal**. To validate this, we conducted experiments on SD3. In the results, **"not split"** refers to using ParaStep without splitting the text encoder, while **"split"** means ParaStep with encoder splitting.
>
> | SD3            | Latency(s) | Memory usage per GPU(MB) |
> | -------------- | ---------- | ------------------------ |
> | p=1            | 18.81      | 16764                    |
> | p=2, not split | 11.16      | 16764                    |
> | p=2, split     | 11.20      | 11386                    |
> | p=4, not split | 6.79       | 16764                    |
> | p=4, split     | 6.84       | 8698                     |
>
> **Q1: Can the selection of key hyperparameters be automated or made adaptive across different diffusion models?**
>
> **Reply:** Thank you for question. We answer this queston in **W2**.
>
> **Q2: For single-GPU scenarios, how does ParaStep (or BatchStep) compare to step-reduction methods?**
>
> **Reply:** Thank you for your question. We will answer this question in two parts, verifying the effects of **ParaStep** and **BatchStep** separately.
>
> As discussed in **W1**, ParaStep is compatible with deterministic schedulers such as DDIM. Regarding the **consistency model**, it is indeed an effective approach for reducing the number of sampling steps in diffusion models. However, it typically requires **costly and often unstable training**, which limits its practicality in some scenarios. In contrast, our method is **training-free**. This makes it broadly applicable across various diffusion models.
>
> To evaluate the efficiency of **ParaStep** in comparison with step-reduction methods, we conduct experiments on **Flux1.dev**, a sota diffusion model for image generation. Flux1.dev employs **FlowMatchEulerDiscreteScheduler**, a deterministic scheduler similar to DDIM. In the Direct Reuse method, the stride *s* indicates that *s - 1* steps are skipped for every *s* steps. We also include comparisons with the original Flux model using a reduced number of sampling steps. As shown in the table below, ParaStep consistently achieves higher generation quality than both the corresponding Direct Reuse configurations and the reduced-step method. These results demonstrate the effectiveness of ParaStep relative to step-reduction methods, and further highlight its robustness even in **extreme few-step** settings.
>
> |                         | Latency(s)↓ | FID↓       | PSNR↑     | SSIM↑      |
> | ----------------------- | ----------- | ---------- | --------- | ---------- |
> | Flux(T=10)              | 5.79        | -          | -         | -          |
> | **Group 1**             |             |            |           |            |
> | Direct Reuse(T=10, s=2) | 3.33        | 157.34     | 17.56     | 0.6427     |
> | Flux(reduced-step, T=5) | 3.30        | 157.44     | 17.56     | 0.6421     |
> | ParaStep(T=10, p=2)     | 3.47        | **141.28** | **18.72** | **0.6751** |
> | **Group 2**             |             |            |           |            |
> | Direct Reuse(T=10, s=4) | 2.50        | 291.83     | 15.69     | 0.5849     |
> | Flux(reduced-step, T=3) | 2.24        | 268.41     | 15.91     | 0.5877     |
> | ParaStep(T=10, p=4)     | 2.60        | 256.70     | 17.36     | 0.6352     |
>
> To evaluate the efficiency of **BatchStep** in comparison with step-reduction method under a single-GPU setting, we conduct experiments on AudioLDM2-base. BatchStep achieves higher generation quality than the step-reduction, demonstrating its effectiveness in single-device scenarios.
>
> |                                | Latency(s)↓ | Speedup↑ | FAD↓   |
> | ------------------------------ | ----------- | -------- | ------ |
> | AudioLDM2(T=200)               | 7.93        | 1        | 1.7765 |
> | AudioLDM2(reduced-step, T=100) | 4.08        | 1.95     | 1.8082 |
> | AudioLDM2(reduced-step, T=50)  | 2.06        | 3.85     | 1.9661 |
> | BatchStep(T=200, s=2)          | 4.30        | 1.84     | 1.7777 |
> | BatchStep(T=200, s=4)          | 2.21        | 3.58     | 1.7805 |
> | BatchStep(T=200, s=8)          | **1.28**    | **6.17** | 1.7897 |
>
> **Q3: Can ParaStep be effectively combined with single-device acceleration methods (such as DeepCache or DDIM) for further speedup?**
>
> **Reply:** Thank you for your suggestion regarding a hybrid approach that combines ParaStep with single-device acceleration methods. Since the steps in the denoising process are **not equally important**, we can apply ParaStep on the more critical steps, and use direct reuse methods (or DeepCache) on the less important ones. We believe this hybrid approach can achieve greater acceleration than ParaStep alone, while maintaining better generation quality than Direct Reuse or DeepCache.
>
> One challenge of implementing this hybrid approach is the need for **dynamic parallelism** (as discussed in Section 4.4 of our paper). ParaStep is inherently a parallel method. For example, if we execute ParaStep with a parallelism degree *p* = 4 for the first 4 steps, and then switch to DeepCache on steps 5 and 6, only one device is required for those steps—leaving the other three devices idle. To make this hybrid method practical and efficient, we would need to develop a system that supports dynamic parallelism, allowing idle devices to serve other incoming requests and thus maximize overall utilization. This is an interesting direction that we believe is worth exploring in future work.

---

> > ### Comment · Reviewer_gLa1 · 2025-08-04
> >
> > Dear Authors,
> >
> > Thanks for your detailed rebuttal!
> >
> > I think most of my concerns have been addressed.
> >
> > Even though I believe it is better to have a better analysis or a systematic solution to define the hyper-parameters within this proposed pipeline together with the some of the uncertainty about the pipeline scalability towards larger diffusion models (due to the requirements of maintaining the entire denoising backbone per device), the optimization idea itself makes sense to me and I would willing to increase my score from 3 to 4 right now.

---

> > > ### Author Response · Authors · 2025-08-05
> > > **Response to Reviewer gLa1**
> > >
> > > Thank you very much for rasing the score! We sincerely appreciate your insightful and constructive comments and suggestions, which have been highly valuable in improving our paper. We will carefully revise the manuscript based on your feedback.

---

### Official Review · Reviewer_W87m · 2025-07-03

**Clarity:** 3
**Significance:** 3
**Originality:** 3
**Rating:** 4
**Confidence:** 5

**Summary:**

This paper proposed a novel parallelization method for communication-efficient multi-GPU diffusion, ParaStep, which periodically distributes the dense computation of different diffusion timestep to different GPU devices. In each cycle, the $i$-th device is responsible for the noise prediction of the $i$-th step in the cycle, while the noise predictions at preceding timesteps are reused from the previous cycle, and the noise predictions at subsequent timesteps are either retrieved from the corresponding devices for the first device or skipped for the rest devices. Extensive experiments on different models and datasets demonstrate the effectiveness of the proposed method.

**Questions:**

See Weaknesses.

**Ethical Concerns:**

["NO or VERY MINOR ethics concerns only"]

**Final Justification:**

The author's rebuttal has satisfactorily addressed my concerns. I believe this work presents a compelling paradigm for seamlessly combining reuse and parallel denoising to advance the trade-off between quality and efficiency. Although the theoretical analysis could be further refined and generalized, I recommend acceptance.

**Limitations:**

yes

**Paper Formatting Concerns:**

No concerns.

**Quality:**

3

**Strengths And Weaknesses:**

Strengths:
1. The paper is well written and easy to understand.
2. The figures are well plotted to help understand the workload of ParaStep.
3. The proposed parallelization method looks interesting to me.

Weaknesses:
1. The main concern is about the theoretical analysis of the proposed method. For example, for different parallelization degrees, what are their upper-bounds of the error distribution induced by reuse-then-prediction?
2. The workload of ParaStep is based on the similarity of noise predictions at adjacent timesteps, which involves reusing the previously predicted noise to update the next-step noisy sample. That is, the authors assume that some timesteps can directly reuse the previous noise to approximate the current step’s without significant deviation. Then, why not adopt direct reuse at some timesteps that may not need further prediction, which can further speedup the inference and reduce the communication overhead.
3.	Most experiments are conducted on the 50-step setting. What about the few-step diffusion setting, e.g., 10-step or less?
4.	Some related works are missing. For example,

[1] Training-Free Adaptive Diffusion with Bounded Difference Approximation Strategy, NeurIPS 2024.

[2] Accelerating Diffusion Transformers with Token-wise Feature Caching, ICLR 2025.

---

> ### Author Rebuttal · Authors · 2025-07-28
>
> Dear reviewer W87m,
>
> Thank you for providing constructive comments and valuable suggestions. Below, we address each point raised.
>
> **W1: The main concern is about the theoretical analysis of the proposed method.**
>
> **Reply:** Thank for your suggestion about about the theoretical analysis. We derive the upper bound of the error distribution introduced by the reuse-then-predict mechanism under a parallelism degree of 2. Specifically, we consider two devices, GPU0 and GPU1. We follow the notations and assumptions established in [1], as well as its derivation for the error bound associated with reuse. We appreciate the solid theoretical foundation provided by [1].
>
> [1] Training-Free Adaptive Diffusion with Bounded Difference Approximation Strategy, NeurIPS 2024.
>
> **Assumptions：**
>
> + $\epsilon_\theta(x, t)$ is Lipschitz w.r.t to its parameters $x$ and $t$
> + The 1st-order difference $\Delta x_i = x_i - x_{i+1}$ exists and is continuous for $0 \leq i \leq T - 1$
>
> **Notation:**
>
> + Let $x$ denote the accurate (ground-truth) sample.
> + Let $\hat{x}$ denote the sample obtained via reuse (referred to as *skip* in [1]).
> + Let $\tilde{x}$ denote the sample obtained via the reuse-then-predict mechanism.
>
> **The denoising step at iteration $i$ is defined as:**
>
> $$x_i = f(i) \cdot x_{i+1} - g(i) \cdot \epsilon_\theta(x_{i+1}, t_{i+1})$$
>
> **The error bound for reusing the sample at iteration $i{-}1$ is derived in [1] as follows: （equation(1)）**
> $$ ||ERR_{i-1}|| = ||\hat{x} _ {i-1}-x_{i-1}||
> \leq  \mathcal{O}(t_i - t_{i+1}) + \mathcal{O}(x_i - x_{i+1}).  \ \ (1)$$
>
>
> Our method is based on the **reuse-then-predict** strategy. Specifically, we assume that reuse is performed at iteration $i{-}1$, followed by a prediction-based correction at iteration $i{-}2$ (See Figure 3(c) for an Intuitive illustration).
>
> **Step $i$: Standard denoising:**
>
> $$x_i = f(i) \cdot x_{i+1} - g(i) \cdot \epsilon_\theta(x_{i+1}, t_{i+1})$$
>
> **Step $i{-}1$: Reuse (executed on GPU1):**
>
> $$\hat{x} _ {i-1} = f(i - 1) \cdot x_i - g(i - 1) \cdot \epsilon_\theta(x_{i+1}, t_{i+1})$$
>
> **Step $i{-}2$: Predict (executed on GPU1):**
>
> $$\tilde{x} _ {i-2} = f(i - 2) \cdot x _ {i-1} - g(i - 2) \cdot \epsilon _ \theta(\hat{x} _ {i-1}, t _ {i-1})$$
>
> + The term $f(i{-}2) \cdot x_{i-1}$ appears here because GPU1 computes $\epsilon_\theta(\hat{x} _ {i-1}, t_{i-1})$ while, **in parallel**, GPU0 has already computed $\epsilon_\theta(x_i, t_i)$. As a result, GPU0 can send $x_{i-1}$ to GPU1 in time for the prediction step.
>
> **The error introduced by the prediction step is:**
>
> $$\tilde{x} _ {i-2}-x _ {i-2}=-g(i-2)*(\epsilon _ \theta(\hat{x} _ {i-1}, t _ {i-1}) - \epsilon _ \theta({x} _ {i-1}, t _ {i-1})) \ \ (2)$$
>
> By the Lipschitz continuity of the assumptions,
>
> $$||\tilde{x} _ {i-2}-x _ {i-2}||<=||g(i-2)*\mathcal{O}(\hat{x} _ {i-1}-x _ {i-1})||=\mathcal{O}(\hat{x} _ {i-1}-x _ {i-1})$$
>
> According to Equation (1) in [1] (error bound for reusing), we can derive the **upper bound of the error distribution introduced by the reuse-then-predict** strategy.
>
> $$||\tilde{x} _ {i-2}-x _ {i-2}||<=\mathcal{O}(\hat{x} _ {i-1}-x _ {i-1}) =\mathcal{O}(t _ i - t _ {i+1}) + \mathcal{O}(x _ i - x _ {i+1})$$
>
> **Why reuse-then-predict leads to lower error in our extensive experiments**
>
> At iteration $i{-}2$, the error introduced by directly applying reuse is:
>
> $$||\hat{x} _ {i-2}-x _ {i-2}||=||g(i-2) \cdot [\epsilon_\theta(x_{i}, t_{i}) - \epsilon_\theta(x_{i-1}, t_{i-1})]||$$
>
> In contrast, the error introduced by reuse-then-predict is:
>
> $$||\tilde{x} _ {i-2}-x _ {i-2}||=||g(i-2)*(\epsilon _ \theta(\hat{x} _ {i-1}, t _ {i-1}) - \epsilon _ \theta({x} _ {i-1}, t _ {i-1}))||$$
>
> The model $\epsilon_\theta$ exhibits a certain degree of **refinement capability**—it can correct errors in the output that stem from inaccuracies in the input. Specifically, it can produce $\epsilon _ \theta(\hat{x} _ {i-1}, t_{i-1})$ that is close to $\epsilon _ \theta(x _ {i-1}, t_{i-1})$, even when $\hat{x} _ {i-1}$ deviates from $x_{i-1}$.
>
> This assumption is supported by Figure 2(b) in the main paper: both reuse and reuse-then-predict exhibit a zigzag pattern in error accumulation—errors rise and then fall periodically. However, the error increments for reuse-then-predict are significantly smaller than those of reuse alone, demonstrating the model’s ability to correct for prior inaccuracies. **This error correction capability explains why reuse-then-predict consistently yields lower overall error in our extensive experiments**.
>
>
>
> **W2: Why not adopt direct reuse at some timesteps that may not need further prediction, which can further speedup the inference and reduce the communication overhead.**
>
> **Reply:** Thank you for your insightful suggestion regarding the potential of applying direct reuse at certain timesteps that may not require further prediction. We agree that this strategy could bring additional speedup.
>
> While direct reuse can indeed accelerate inference, it often leads to a noticeable drop in generation quality. The core contribution of our ParaStep method lies in the proposed **reuse-then-predict** mechanism, which explicitly refine the quality degradation caused by direct reuse. Since steps in the denoising process **are not equally important**, we believe a hybrid approach—applying ParaStep at critical steps and direct reuse at less important ones—could achieve greater acceleration than ParaStep alone, while maintaining better generation quality than pure direct reuse.
>
> One challenge of implementing this hybrid strategy lies in its dynamic nature. ParaStep is designed as a parallel method with a fixed degree of parallelism. Introducing dynamic parallelism (as discussed in Section 4.4 of our paper) requires the ability to reallocate computing resources across steps. For instance, if we apply ParaStep with a parallelism degree *P* = 4 for the first 4 steps, and then apply direct reuse at steps 5 and 6, only one device is needed for those steps, leaving the remaining three devices idle. To make such a hybrid approach practical, we would need to design a system that supports dynamic parallelism—allowing idle devices to be reassigned to other requests during execution, thereby maximizing resource utilization. This is an interesting direction that we believe is worth exploring in future work.
>
>
>
> **W3: Most experiments are conducted on the 50-step setting. What about the few-step diffusion setting, e.g., 10-step or less?**
>
> **Reply:** Thank you for your valuable comment. Most of our experiments are conducted under the 50-step setting, as this is the default configuration for models such as CogVideoX-2B and Latte. Setting the number of steps of CogVideoX-2B to 35 results in a significant drop in generation quality.
>
> |                     | Latency(s)↓ | lpips↓     | PSNR↑       | SSIM↑      |
> | ------------------- | ----------- | ---------- | ----------- | ---------- |
> | CogVideoX-2B(T=50)  | 86.92       | -          | -           | -          |
> | CogVideoX-2B(T=40)  | 70.94       | 0.3102     | 18.4771     | 0.6931     |
> | CogVideoX-2B(T=35)  | 64.12       | 0.4033     | 16.2626     | 0.6403     |
> | ParaStep(T-50, p=2) | **58.24**   | **0.0247** | **35.2605** | **0.9588** |
>
> To evaluate the performance of ParaStep under few-step settings, we additionally conducted experiments on Flux1.dev, which supports few-step generation. We compare the outputs generated by various acceleration methods (e.g., Direct Reuse, reduced-step generation, and ParaStep) against those from the original model (Flux with *T* = 10) to evaluate the generation quality. Here, *P* denotes the degree of parallelism in ParaStep, and *s* represents the stride used in the Direct Reuse method.
>
> |                         | Latency(s)↓ | FID↓       | PSNR↑     | SSIM↑      |
> | ----------------------- | ----------- | ---------- | --------- | ---------- |
> | Flux(T=10)              | 5.79        | -          | -         | -          |
> | **Group 1**             |             |            |           |            |
> | Direct Reuse(T=10, s=2) | 3.33        | 157.34     | 17.56     | 0.6427     |
> | Flux(reduced-step, T=5) | 3.30        | 157.44     | 17.56     | 0.6421     |
> | ParaStep(T=10, p=2)     | 3.47        | **141.28** | **18.72** | **0.6751** |
> | **Group 2**             |             |            |           |            |
> | Direct Reuse(T=10, s=4) | 2.50        | 291.83     | 15.69     | 0.5849     |
> | Flux(reduced-step, T=3) | 2.24        | 268.41     | 15.91     | 0.5877     |
> | ParaStep(T=10, p=4)     | 2.60        | 256.70     | 17.36     | 0.6352     |
>
> The results in the table demonstrate that ParaStep achieves a clear speedup even under few-step settings, while also outperforming both Direct Reuse and reduced-step generation in terms of output quality.
>
>
>
> **W4: Some related works are missing.**
>
>  **Reply:** Thank you for your valuable suggestion on citing [1] and [2], both of which are indeed closely related to our work. We will cite them in the revised version. Specifically, [1] employs a skipping strategy guided by third-order latent differences, aiming to skip as many noise prediction steps as possible. [2] introduces token-wise feature caching, enabling adaptive selection of the most suitable tokens for caching to achieve acceleration. Both of these studies, as well as our work, are inspired by the concept of **reuse**. The key difference is that [1] [2] adopts direct reuse while we propose and adopt a reuse-then-predict mechanism, which effectively mitigates the quality degradation typically caused by direct reuse.

---

> > ### Comment · Reviewer_W87m · 2025-08-04
> >
> > Thank you for the rebuttal—it has satisfactorily addressed my concerns. I believe this work presents a compelling paradigm for seamlessly combining reuse and parallel denoising to advance the trade-off between quality and efficiency. Although the theoretical analysis could be further refined and generalized, I recommend acceptance and will maintain my score of 4.

---

> > > ### Author Response · Authors · 2025-08-05
> > > **Response to Reviewer W87m**
> > >
> > > Thank you very much for your recognition of our work. We sincerely appreciate your insightful and constructive comments and suggestions, which have been highly valuable in improving our paper. We will carefully revise the manuscript based on your feedback.

---

### Official Review · Reviewer_p6GU · 2025-07-03

**Clarity:** 3
**Significance:** 3
**Originality:** 3
**Rating:** 4
**Confidence:** 4

**Summary:**

The paper tackles the high inference latency of diffusion models, which arises from the sequential nature of the denoising process. The authors identify that existing parallelization strategies often suffer from high communication overhead, limiting their use to expensive, high-bandwidth data center hardware.

To address this, the paper makes the following key contributions:
- It introduces the reuse-then-predict mechanism. This is the core novelty. Based on the observation that predicted noise is similar across adjacent timesteps, the method first reuses the noise from a previous step to generate an approximate next latent state, and then predicts a refined noise from this new latent. This avoids the quality degradation of "direct reuse".
- It proposes a distributed algorithm that leverages this mechanism to parallelize the computation of multiple denoising steps across several GPUs. This design requires only lightweight, step-wise communication, making it suitable for commodity hardware.
- It introduces a single-device variant for non-compute-intensive models (like audio), which transforms the parallel computation into a more efficient batched execution on one GPU.
- It provides extensive experimental results on state-of-the-art image (SD3), video (SVD, CogVideoX, etc), and audio (AudioLDM2) models, showing significant speedups over baselines with minimal quality degradation.

**Questions:**

Please refer to the weaknesses.

**Ethical Concerns:**

["NO or VERY MINOR ethics concerns only"]

**Final Justification:**

The rebuttal addressed my concerns, I therefore decided to raise my score.

**Limitations:**

Please refer to the weaknesses.

**Quality:**

3

**Strengths And Weaknesses:**

**Strengths:**

1.  The `reuse-then-predict` idea is novel, and its focus on low communication overhead makes ParaStep a highly practical solution for accelerating diffusion models on widely available hardware.
2.  he method is shown to be effective across multiple modalities (image, video, audio) and consistently outperforms strong, recent baselines in terms of speedup while maintaining competitive or superior generation quality.
3.  The inclusion of `BatchStep` for single-GPU acceleration and the demonstration of compatibility with `TeaCache` enhance the paper's contribution.

**Weaknesses:**

1. ParaStep requires replicating the entire model on each GPU. This is a significant limitation that constrains its applicability to models that can fit into the memory of a single GPU. While mentioned briefly in the Appendix, this critical trade-off (trading memory for speed) deserves a more prominent discussion in the main body of the paper.
2.  As mentioned above, the paper lacks a detailed performance breakdown (computation vs. communication). Such an analysis is crucial to definitively prove that the speedup comes from communication efficiency and to better understand the performance characteristics of both the proposed method and the baselines.
3. Fig. 3 is slightly complex. It is suggested that the author could hide the technical details and show more of the high level idea.

---

> ### Author Rebuttal · Authors · 2025-07-27
>
> Dear reviewer p6GU,
>
> Thank you for providing constructive comments and valuable suggestions. Below, we address each point raised.
>
> **W1: ParaStep requires replicating the entire model on each GPU. This is a significant limitation that constrains its applicability to models that can fit into the memory of a single GPU.**
>
> **Reply:** Thank you for your valuable comment. We apologize for this unclear description in Section Limitations. A typical diffusion pipeline consists of several components, including a text encoder, a noise predictor (e.g., DiT or U-Net), and a VAE, among others. The model as mentioned in the paper denotes the noise predictor, and ParaStep only requires replicating the entire noise predictor across all devices. However, the text encoder, typically the primary memory bottleneck, is not replicated. Instead we **split the text encoder into multiple stages across devices** to reduce per-device memory consumption. We will clarify this point more explicitly in the revised version of the paper. The table below presents the memory breakdown for SD3, CogVideoX-2B, and Latte.
>
> | Pipeline     | Total(MB) | Text encoder(MB) | Noise predictor(MB) |
> | ------------ | --------- | ---------------- | ------------------- |
> | SD3          | 16764     | 11004            | 4024                |
> | CogVdieoX-2b | 14769     | 11004            | 3342                |
> | Latte        | 13188     | 11004            | 2019                |
>
> A natural concern regarding splitting the text encoder is whether this would increase the overall pipeline latency. Since the text encoder is only invoked once to process the input prompt, and splitting it only introduces a single communication step between adjacent stages, the additional latency incurred is minimal. To validate this, we conducted experiments on SD3 to get the latency. In the results, **"not split"** refers to using ParaStep without splitting the text encoder, while **"split"** refers to using ParaStep with a split text encoder. *P* denotes the degree of parallelism. As shown, splitting the text encoder significantly reduces memory usage, while incurring only a minor increase in latency. By applying both **ParaStep** and **text encoder splitting**, we achieve not only lower latency but also reduced memory usage compared to the baseline.
>
> | SD3            | Latency(s) | Memory usage per GPU(MB) |
> | -------------- | ---------- | ------------------------ |
> | p=1            | 18.81      | 16764                    |
> | p=2, not split | 11.16      | 16764                    |
> | p=2, split     | 11.20      | 11386                    |
> | p=4, not split | 6.79       | 16764                    |
> | p=4, split     | 6.84       | 8698                     |
>
>
>
> **W2: As mentioned above, the paper lacks a detailed performance breakdown (computation vs. communication). Such an analysis is crucial to definitively prove that the speedup comes from communication efficiency and to better understand the performance characteristics of both the proposed method and the baselines.**
>
> **Reply:** Thank you for your valuable suggestion regarding a more detailed performance breakdown. To demonstrate that the observed speedup primarily stems from improved communication efficiency, we conducted experiments on the SVD model. Specifically, we measured both the total latency and the communication-related latency for AsyncDiff and our proposed ParaStep. As shown in the table below, ParaStep exhibits significantly lower communication latency compared to AsyncDiff, which directly contributes to its higher overall speedup.
>
> |                | Total Latency(s) | Latency of computation(s) | Latency of communication(s) |
> | -------------- | ---------------- | ------------------------- | --------------------------- |
> | SVD            | 51.04            | 51.04                     | -                           |
> | AsyncDiff(p=2) | 39.30            | 30.71                     | 8.59                        |
> | ParaStep(p=2)  | 30.52            | 30.42                     | 0.10                        |
> | AsyncDiff(p=4) | 31.30            | 21.01                     | 10.29                       |
> | ParaStep(p=4)  | **20.08**        | 19.94                     | 0.14                        |
>
> We also provide the performance breakdown on CogVideoX-2B and Latte. Since AsyncDiff does not currently support these models, we report the breakdown results for ParaStep only.
>
> |               | Total Latency(s) | Latency of computation(s) | Latency of communication(s) |
> | :-----------: | ---------------- | ------------------------- | --------------------------- |
> | CogVideoX-2b  | 91.69            | 91.69                     | -                           |
> | ParaStep(p=2) | 51.32            | 50.98                     | 0.34                        |
> | ParaStep(p=4) | 34.85            | 34.22                     | 0.63                        |
>
> |               | Total Latency(s) | Latency of computation(s) | Latency of communication(s) |
> | :-----------: | ---------------- | ------------------------- | --------------------------- |
> |     Latte     | 32.83            | 32.83                     | -                           |
> | ParaStep(p=2) | 20.25            | 20.03                     | 0.22                        |
> | ParaStep(p=4) | 13.38            | 12.92                     | 0.46                        |
>
>
>
> **W3: Fig. 3 is slightly complex. It is suggested that the author could hide the technical details and show more of the high level idea.**
>
> **Reply:** Thank you for your valuable suggestion. We will simplify the Fig.3 by hiding lower-level technical details to highlight the overall design and the high level idea in the revised version of the paper.

---

> > ### Author Response · Authors · 2025-08-06
> > **Sincerely looking forward to your feedback**
> >
> > Dear Reviewer p6GU,
> >
> > Thank you for your insightful and constructive comments, which have been very helpful in improving our paper. We have made our best effort to address all of your concerns. As the discussion phase deadline is approaching, we would greatly appreciate it if you could let us know whether our responses have adequately addressed your concerns. We would be happy to provide further clarification if there are any remaining issues.

---

> > ### Comment · Reviewer_p6GU · 2025-08-07
> >
> > Thanks for the reply and clarification. The additional experiments and data provided a satisfactory response to my questions. The rebuttal addressed my concerns, and I decided to raise my score.

---

> > > ### Author Response · Authors · 2025-08-08
> > > **Response to Reviewer p6GU**
> > >
> > > Thank you very much for rasing the score! We sincerely appreciate your insightful and constructive comments and suggestions, which have been highly valuable in improving our paper. We will carefully revise the manuscript based on your feedback.

---

> ### Comment · Area_Chair_jzgm · 2025-08-05
>
> Dear Reviewer,
>
> We are approaching the end of the discussion period with the authors and the authors have provided you with their rebuttal.
>
> May I please kindly ask to you check on the author rebuttal and engage in a meaningful discussion with authors?
>
> There is also a new rule this year: you will need to tick the mandatory acknowledgement box, but **only after discussing with the authors**.
>
> Thanks.
> AC

---

### Decision · Program_Chairs · 2025-09-17

**Decision:**

Accept (poster)

**Comment:**

This paper presents a new parallelization strategy for diffusion inference and demonstrates significant speedups across various models. Although its score is still at boaderline, after discussions/rebuttal with the reviewers, several concerns have been addressed, primarily regarding experimentation (e.g., limited time-steps, detailed performance breakdowns). I believe these improvements have strengthened the paper, and I would therefore recommend Acceptance as a poster.